# Non-Categorical Analyses Identify Rotenone-Induced ‘Parkinsonian’ Rats Benefiting from Nano-Emulsified Punicic Acid (Nano-PSO) in a Phenotypically Diverse Population: Implications for Translational Neurodegenerative Therapies

**DOI:** 10.3390/ijms252312635

**Published:** 2024-11-25

**Authors:** Jennifer Viridiana Sánchez-Camacho, Margarita Gómez-Chavarín, Nuria Galindo-Solano, Patricia Padilla-Cortés, José Luis Maldonado-García, Gilberto Pérez-Sánchez, Lenin Pavón, Jesús Ramírez-Santos, Gabriel Roldán Roldán, Modesto Gómez-López, Gabriel Gutierrez-Ospina

**Affiliations:** 1Sección de Estudios de Posgrado, Escuela Superior de Medicina, Instituto Politécnico Nacional, Ciudad de México 11340, Mexico; moygl@yahoo.com.mx; 2Laboratorio de Medicina Regenerativa y Canales Iónicos, Departamento de Fisiología, Facultad de Medicina, Universidad Nacional Autónoma de México, Ciudad de México 04510, Mexico; margaritachavarin@gmail.com; 3Laboratorio de Biología de Sistemas, Departamento de Biología Celular y Fisiología, Instituto de Investigaciones Biomédicas, Universidad Nacional Autónoma de México, Ciudad de México 04510, Mexico; nuria.gs@ciencias.unam.mx (N.G.-S.); ramirezs@iibiomedicas.unam.mx (J.R.-S.); 4Programa de Doctorado en Ciencias Biomédicas, Unidad de Posgrado, Universidad Nacional Autónoma de México, Ciudad de México 04510, Mexico; 5Unidad de Cromatografía, Instituto de Investigaciones Biomédicas, Universidad Nacional Autónoma de México, Ciudad de México 04510, Mexico; ppadillac@iibiomedicas.unam.mx; 6Departamento de Bioquímica, Facultad de Medicina, Universidad Nacional Autónoma de México, Ciudad de México 04510, Mexico; joselmgarci@comunidad.unam.mx; 7Departamento de Inmunología, Escuela Nacional de Ciencias Biológicas, Instituto Politécnico Nacional, Ciudad de México 11340, Mexico; 8Laboratorio de Psicoinmunología, Dirección de Investigaciones en Neurociencias, Instituto Nacional de Psiquiatría “Ramón de la Fuente Muñíz”, Ciudad de México 14370, Mexico; 9Laboratorio de Neurología Conductual, Departamento de Fisiología, Facultad de Medicina, Universidad Nacional Autónoma de México, Ciudad de México 04510, Mexico; gabaergico@gmail.com; 10Coordinación de Psicobiología y Neurociencias, Facultad de Psicología, Universidad Nacional Autónoma de México, Ciudad de México 04510, Mexico

**Keywords:** translational science, categorical thinking, systematic heterogenization in animal research, phenotype variance, pomegranate, conjugated linolenic acid, Parkinson’s disease

## Abstract

The pursuit of nutraceuticals to improve the quality of life for patients with neurodegenerative conditions is a dynamic field within neuropharmacology. Unfortunately, many nutraceuticals that show promise in preclinical studies fail to demonstrate significant clinical benefits in human trials, leading to their exclusion as therapeutic options. This discrepancy may stem from the categorical interpretation of preclinical and clinical results. Basic researchers often assume that non-human experimental animals exhibit less phenotypic variability than humans. This belief overlooks interindividual phenotype variation, thereby leading to categorical conclusions being drawn from experiments. Consequently, when human clinical trials are conducted, the researchers expect similarly conclusive results. If these results are not achieved, the nutraceutical is deemed ineffective for clinical use, even if numerous individuals might benefit. In our study, we evaluated whether analyzing phenotype variability and similarity through non-categorical methods could help identify rotenone (ROT)-treated rats that might benefit from consuming nano-emulsified punicic acid (Nano-PSO), even if the prevention of “parkinsonism” or the restoration of neurometabolic function is inconsistent across individuals. Our findings supported this hypothesis. The benefits of Nano-PSO were not categorical; however, analyzing phenotype variance allowed us to identify ROT rats with varying degrees of benefit from Nano-PSO consumption. Hence, the translational potential of results from basic science studies testing nutraceuticals as pharmaceutical products against neurodegeneration may improve if researchers also interpret their results using non-categorical methods of data analysis for population screening, even if the overall therapeutic outcomes for the entire population show internal inconsistencies.

## 1. Introduction

An active area of research in neuropharmacology seeks to identify natural compounds that can be used to treat common neurodegenerative diseases. Unfortunately, many nutraceuticals that show promise in preclinical studies fail to demonstrate efficacy in human clinical trials [1,2,3,4,5]. While several factors may contribute to this discrepancy, we think that the bias introduced by the categorical interpretation of results obtained in preclinical and clinical studies plays a significant role. This is not surprising, as categorical thinking consistently underestimates the value and biological meaning of phenotypic variation [6,7,8]. ***By overlooking phenotypic variance, categorical interpretations prevent the identification of individuals who could benefit from the treatment, even if the therapeutic effect is inconsistent across the population*.** Recognizing this limitation is of the utmost importance, as numerous patients undergoing neurodegeneration or those with a life-long susceptibility to developing it [9] may have diverse phenotypes [10,11,12,13,14,15,16,17,18] that could respond differently to nutraceuticals. This might be the case for the Nano-PSO, as patients with neurodegenerative diseases often exhibit a range of dysfunctional mitochondrial phenotypes [19,20,21,22,23]. Some of these phenotypes could potentially respond to Nano-PSO, due to its capacity to enhance mitochondrial function [24].

To explore the validity of the premise set out in the preceding paragraph (highlighted in **bold** and *italics*), we used a non-categorical analysis to assess phenotype variance in a group of rats administered ROT. Some rats did not receive treatment, while others were treated with Nano-PSO either before or after ROT administration. ROT is reported to *categorically* produce dyskinesias and dopaminergic neuronal degeneration in the midbrain of rodents [3,25] by uncoupling complex I from the mitochondrial respiratory chain [26,27]. In contrast, Nano-PSO is said to *categorically* neuro-protect or neuro-restore neurological functional morphology in mouse models of Creutzfeldt–Jakob disease (n = 4 TgMHu2ME199K mice [28]; n = 6–10 TgMHu2ME199K mice [29]), multiple sclerosis (n = 6–8 female C57BL/6 mice), or Alzheimer’s disease (n = 5–18 5xFAD × C57BL/6 mice [24]) [24,28,29,30,31]. It is also claimed that Nano-PSO *categorically* reduces cognitive decline in patients with multiple sclerosis (Group A: n = 12; Group B: n = 10) [31]. However, in these studies, the central tendency values are insufficient to predict the outcomes for all the subjects studied. In fact, their results show (1) different individuals exhibiting the disease phenotype differently, (2) everyone responding to Nano-PSO to varying degrees, and (3) the treatment influencing the parameters used to monitor improvement differently within and between experimental groups. These are examples of studies, among hundreds of others, that have drawn misleading categorical conclusions.

Building on the previous arguments, this study evaluated whether analyzing phenotype variability through non-categorical methods of data analysis could help identify rats that might benefit from consuming Nano-PSO after being administered ROT, even if the benefits were not categorically observed in the treated population. Our results show that “trait rotenonization” was highly variable, as were the benefits of consuming Nano-PSO. Despite variations, the analysis of phenotype variance and similarity allowed us to identify ROT-treated rats with varying degrees of benefit from consuming Nano-PSO, whether it was used as a neuroprotective or neurorestorative agent. Therefore, we propose that the translational success of basic science studies investigating nutraceuticals as treatments for neurodegeneration could be enhanced by employing proper patient screening and selection using non-categorical methods of data analysis.

## 2. Results

### 2.1. General Considerations

A prevailing assumption in biomedical experimental science is that the phenotypes of animals commonly used to model medical conditions are relatively invariant. However, categorical thinking tends to “*amplify differences across category boundaries, assuming that individuals in different categories are more different than they are, and … compress within categories, assuming that individuals in the same category are more similar than they are*” [7]. These cognitive biases lead to false polarization, a dilemma that undermines the possibility of translating the benefits of therapeutic agents to the population when the results of clinical studies are not as robust as expected, based on the “categorical” claims made by basic researchers. The categorical perspective, however, overlooks that developmental trajectories exhibit remarkable individual diversity and that as organisms mature, their phenotypic expressions weave a complex tapestry of variation [6]. To address this conflict, forward-thinking researchers advocate introducing the “*systematic heterogenization of study samples and conditions by actively incorporating biological variation into the study design through diversification of study samples and conditions*” [32]. In this context, qualitative comparative analysis [33] can not only improve reproducibility but also unlock the translation potential of basic research findings into real-world clinical settings, even when the results obtained in the population studied, whether of experimental animals or patients, appear internally inconsistent.

Considering the aforementioned context, in this study, we analyzed the variation and similarity of phenotypes using principal component analysis (PCA) and hierarchical clustering dendrogram analysis (HCDA). Our objective was to identify rats that might benefit from the consumption of Nano-PSO, used as a neuroprotective or neurorestorative agent, when it was administered before or after receiving ROT, even if such benefits were inconsistently observed in a heterogeneous population of rats. For our statistical analyses, we considered “rotenonization” as an individually variable intrinsic trait and considered therapeutic responses to Nano-PSO as a variable but state-dependent trait. To facilitate interindividual comparisons, given the technical impossibility of performing a longitudinal study, we unnaturally created a series of phenotypes based on quantifiable parameters. Each phenotype was defined based on the statistical interaction between the type of treatment each rat received and all the parameters specific to the phenotype in question. In all cases, we considered rats that were administered ROT and supplemented with Nano-PSO as relatively neuro-protected or neuro-restored if their phenotypic variability and similarity clustered them closer to the phenotypes of intact rats. Finally, for our analysis, the phenotype of each rat was defined using as many parameters as possible, depending on the sampling restrictions associated with each analytical technique used. In some cases, different samples of the same rat were processed to gather data through different techniques; in others, this was not possible. The phenotypes and the parameters used to define them are listed below:(A)The motor phenotype (MoP), defined by the statistical interaction between the treatment and the latency values obtained in each of the assessments conducted during the inclined beam test. These assessments were performed 21 days and 42 days after the initiation of treatment.(B)The morphotype (MrP), defined by the statistical interaction between the type of treatment and the number of dopaminergic neurons present in the right side or left side and in the combined total within the *Substantia nigra pars compacta* (*SNpc*). The morphological assessment was conducted in fixed brains collected 42 days after the initiation of treatment.(C)The α-synuclein phenotype (αSP), defined by the statistical interaction between the treatment and the concentration of α-synuclein (α-syn), as measured in homogenates of the caudate nucleus (*CaNu*) and (*SNpc*). Tissue samples were collected bilaterally on day 42 following the initiation of treatment.(D)The dopaminergic phenotype (DP [34]), defined by the statistical interaction between the treatment and the concentration of dopamine (DA), 3,4-dihydroxyphenylacetic acid (DOPAC), the DA:DOPAC ratio, and serotonin (5HT), as assessed in the homogenates of *CaNu* and *SNpc*. Tissue samples were collected bilaterally on day 42 following the initiation of treatment.(E)Lipid peroxidation phenotype (LPP), defined by the statistical interaction between the treatment and the concentration of malondialdehyde (MDA) in homogenates of *CaNu* and *SNpc*. Tissue samples were collected bilaterally on day 42 following the initiation of treatment.(F)Transcriptomic phenotype (TP), defined by the statistical interaction between the treatment and the relative transcription levels of genes encoding tyrosine hydroxylase (TH), α-syn, glucose transporter 3 (GLUT3), glucose transporter 4 (GLUT4), catalase (CAT), glutathione peroxidase 1, (GPX1), and superoxide dismutase (SOD) in homogenates of *CaNu*. Tissue samples were collected bilaterally on day 42 following the initiation of treatment.(G)Anti-oxidative phenotype (AOP), defined by the statistical interaction between the treatment and the activity levels of CAT, GPx, and SOD, as estimated in blood samples withdrawn 42 days after the initiation of treatment.(H)Glucose, triglycerides, and cholesterol profile (GTCP), defined by the statistical interaction between the treatment and the concentration of glucose, triglycerides, and cholesterol in blood samples withdrawn 42 days after the initiation of treatment.

### 2.2. Assessing MoPs Variance and Similitude

The PCA (Figure 1a) revealed a significant variation in MoP among intact, neuro-protected, and neuro-restored rats, unlike those administered ROT. This indicates that the MoPs of intact, neuro-protected, and neuro-restored rats are non-categorical. Notably, Nano-PSO appears to preserve rather than recover motor performance, as evidenced by the substantial overlap and proximity of the variance ellipses for intact and neuro-protected rats in the PCA dimensional space. The degree of neuroprotection achieved varied significantly among rats, with only some displaying nearly normal MoPs. Interestingly, while intact rats performed similarly across both tests, some neuro-protected and neuro-restored rats showed better performance in the first test, while others improved in the second test, as indicated by the orientation of the ellipses in the PCA dimensional space. Lastly, it is important to note that “rotenonization” decreased MoP variability, suggesting that a healthier population exhibits greater interindividual phenotypic variance; Nano-PSO tends to restore MoP variability.

The HCDA (Figure 1b) was used to estimate the degree of MoP similarity between rats subjected to different experimental treatments. MoP similarity was high among ROT-administered rats, as they clustered closely together. This was not the case for intact, neuro-protected, and neuro-restored rats, which exhibited varying degrees of intermixing depending on their relative positions along the dendritic tree. Overall, neuro-protected rats clustered closer to intact rats than neuro-restored rats. This suggests that the MoPs of neuro-protected rats are more similar to those of intact rats. However, the degree of similarity between neuro-protected and intact rats varied significantly, with the MoPs of neuro-protected rats, coded as R2, R5, R7, and R8, being the closest to those of intact rats. In other words, the neuro-protected rats, R2, R5, R7, and R8, are better candidates for a preventive protocol with Nano-PSO.

### 2.3. Assessing MrPs Variance and Similitude

The PCA (Figure 2a) reveals significant variation in MrPs among intact, neuro-protected, neuro-restored, and ROT-administered rats, indicating that the MrPs in all groups are non-categorical. Notably, Nano-PSO appears to preserve the number of dopaminergic neurons more effectively than recovering them, as evidenced by the proximity of the variance ellipses for intact and neuro-protected rats in the PCA dimensional space. The degree of neuroprotection varies among rats, with none maintaining MrPs close to normal values. Additionally, neuroprotection is not homogeneous; in some rats, it progresses better in the left *SNpc* than in the right, while in others, the opposite occurs. Dopaminergic neurodegeneration also occurs asymmetrically between the sides in the mesencephalon of ROT-administered rats. The anatomical asymmetry of the effects of Nano-PSO and ROT on dopaminergic neuron numbers can be deduced from the orientation of the ellipses in the PCA dimensional space. Our observations suggest that Nano-PSO supplementation maintains pre-existing asymmetries, as a degree of asymmetry in the number of dopaminergic neurons was also observed in intact rats. This finding is clinically relevant because Parkinson’s disease (PD) motor prodromal symptoms are frequently subtle and unilateral [35]. Lastly, “rotenonization” also decreased MrP variability, suggesting again that a healthier population exhibits greater interindividual phenotypic variance. The relative neuroprotective effect of Nano-PSO was, however, insufficient to reinstate MrP variability in the population of neuro-protected rats.

The HCDA (Figure 2b) was utilized to assess the similarity of MrPs among rats subjected to various experimental treatments. Overall, MrPs displayed diversity within each group. Although intact and neuro-protected rats branched early and distinctly from each other and clustered with their peers, neither group formed categorical clusters. The intermingling of neuro-restored and ROT-treated rats in clusters and subclusters indicates that they do not represent categorical phenotypes and that Nano-PSO does not restore dopaminergic neuron numbers. Neuro-protected rats share a common principal branch with neuro-restored and ROT-treated rats but diverge from them earlier in the branching pattern. The combined results suggest that Nano-PSO provides some degree of neuroprotection, although its extent is insufficient to maintain normal dopaminergic neuron numbers. Nonetheless, based on their MrPs, specimens R1-R6 might be considered for inclusion in a Nano-PSO supplementation protocol, even if the benefit is modest. Further segmentation aimed at more specifically targeting specimens likely to respond robustly to Nano-PSO would only include mice R2 and R5, as they display MrP values closer to normal values.

A fundamental finding from our results is that MrPs do not predict MoPs (compare Figure 1b and Figure 2b). In other words, the number of dopaminergic neurons does not predict linearly motor performance, especially in neuro-protected rats. Such inconsistencies have already been reported previously [36,37]. Although compensatory changes may be involved (see below), this observation implies that personalizing treatments by using non-categorical analytical methods would require increasing the number of phenotyping parameters to address a population’s interindividual variance [38].

This is why some authors have suggested including metabolic parameters to better phenotype those patients that are undergoing neurodegeneration before prescribing a specific treatment [36,37].

### 2.4. Assessing αSPs Variance and Similitude

The PCA (Figure 3a) shows the variance of αSP as being highest in rats administered ROT and moderate in neuro-protected and neuro-restored rats. Four distinct subgroups of αSPs emerge after closer examination of the data from neuro-protected and neuro-restored rats. Each subgroup consists of varying proportions of neuro-protected and neuro-restored rats whose αSPs exhibit similar characteristics. Collectively, this data set suggests that αSPs in neuro-protected, neuro-restored, and ROT-administered rats are intrinsically inconsistent. When it comes to intact rats, it appears that their αSPs exhibit reduced variance, leading to behavior that leans toward the categorical. Overall, these results suggest that Nano-PSO offers neuroprotection and neuro-restoration against α-syn accumulation in the *CaNu* and *SNpc*. This conclusion gains support from the proximity of the variance ellipse for neuro-protected and neuro-restored rats to the cluster of intact rats within the PCA dimensional space. Notably, the magnitude of both effects varies individually. Therefore, the response to Nano-PSO is not categorical.

In addition, the effects of ROT toxicity, neuroprotection, and neuro-restoration—monitored through α-syn availability—demonstrate individual regionalization. In specific rats, these effects predominantly unfold in the *SNpc*, while in others, they manifest in the *CaNu*. We infer this regionalization from the orientation of the variance ellipse for neuro-protected and neuro-restored rats within the PCA dimensional space. Interestingly, neuro-protected and neuro-restored rats with increased α-syn availability in the *CaNu* cluster closely to the intact rat group. These specimens appear to benefit the most from Nano-PSO supplementation. One possible explanation for these results is that the αSPs in which α-syn accumulates in the *CaNu* may possess a greater functional reserve [39,40], regardless of whether they accumulate smaller amounts or are distinct conformers of α-syn [41].

The HCDA (Figure 3b) was utilized to estimate the similarity of αSPs among rats subjected to various experimental treatments. The αSPs of neuro-protected and neuro-restored rats exhibited the highest similarity because they are clustered combined. Both groups of Nano-PSO-supplemented rats branched with intact rats, suggesting that Nano-PSO supplementation results in αSPs that are more similar to those of intact rats, although they are still distinct from normal conditions. Conversely, rats administered ROT branched distantly from the other groups; however, their cluster is non-categorical. Finally, the αSPs of intact rats were quite similar; their cluster is intrinsically consistent. These combined results indicate that while Nano-PSO provides some degree of neuroprotection and neuro-restoration, this is insufficient to maintain normal levels of α-syn availability. Despite this outcome, based on the αSPs, the specimens coded as R1 and R4 in the neuro-protected and neuro-restored groups may continue under Nano-PSO supplementation because they receive the most benefit.

### 2.5. Assessing DPs Variance and Similitude

The PCA uncovered the magnitude of DP variance in the *SNpc* (Figure 4a) and *CaNu* (Figure 5a) values of intact, neuro-protected, neuro-restored, and ROT-administered rats. DP variance is notable within groups, thus indicating that they all have low internal consistency. DP variability is greatest in intact rats, followed, in decreasing order, by the populations of neuro-protected, neuro-restored, and ROT-administered rats. Despite this variability, the relative position, dimension, and orientation of the variance ellipses in the PCA dimensional space demonstrate that Nano-PSO conferred the greatest benefits when used as a neuro-restorative. For both *SNpc* and *CaNu* DPs, the extent of neuro-restoration and neuroprotection showed individual variability.

Further analysis indicates that (1) *SNpc* and *CaNu* DPs are not correlated within the same specimen or across different specimens, and (2) rotenone impairs DPs to varying degrees in both brain regions. Additionally, the ratio of DA:DOPAC notably influences the variability of *SNpc* DPs in intact, neuro-protected, and ROT-administered rats. In neuro-restored rats, the variance of *SNpc* DPs is more strongly influenced by the concentration of DA, DOPAC, and 5HT. Conversely, in *CaNu* DPs, the DA: DOPAC ratio is the primary factor driving variability in neuro-protected and ROT-administered rats, while DA and DOPAC concentrations are predominant in neuro-restored rats. In intact rats, the contributions of various factors to *CaNu* DP variability is relatively balanced, with DA concentrations having a slightly greater influence. Notably, the variability of *CaNu* DPs is minimally affected by 5HT concentrations, unlike *SNpc* DPs, in which 5HT plays a more significant role. This finding is important because serotonergic neurons regulate DA release and the concentration of DA and may provide DA under Parkinsonian conditions [42]. Nano-PSO may, therefore, promote the survival of 5HT neurons and/or enhance their compensatory functions, similar to the effects observed with other omega fatty acids [43].

The HCDAs evaluated the similarity of DPs in *SNpc* (Figure 4b) and *CaNu* (Figure 5b) among rats subjected to various experimental treatments. The analyses revealed that all *SNpc* and *CaNu* DPs are dissimilar, both within and between the experimental groups. Although the results suggest that Nano-PSO provides some degree of neuro-restoration and, to a lesser extent, neuroprotection, the level of improvement achieved is insufficient to maintain DPs within the normal range. However, specimens from neuro-restored rats, specifically the *SNpc* DPs coded as R6-R8 and *CaNu* DPs coded as R2, R4, R5, and R6, may benefit from continued Nano-PSO supplementation.

### 2.6. Assessing LPPs Variance and Similitude

Rotenone interferes with mitochondrial function and hence increases lipid peroxidation [44,45]. This is why we initially considered this parameter to be a more precise biomarker; however, this was not the case. The PCA discloses the extent of the LPP variance of intact, neuro-protected, neuro-restored, and ROT-administered rats (Figure 6a). LPP variance is notable within the groups, except in intact rats, indicating that LPPs are intrinsically inconsistent. LPP variability is greatest in ROT-administered rats, followed, in decreasing order, by the populations of neuro-protected, neuro-restored, and intact rats. Despite this variability, the relative position, dimension, and orientation of the variance ellipses in the PCA dimensional space indicate that Nano-PSO provides the greatest benefits when used as a neuroprotectant. The degree of neuro-restoration and neuroprotection was moderate, although it varied among individuals. Notably, LPP variability in neuro-restored rats is predominantly influenced by lipid peroxidation in the *SNpc*. In intact rats, the lipid peroxidation in *CaNu* drives the LPPs’ small variability. In neuro-protected and ROT-treated rats, the contributions of lipid peroxidation in *SNpc* and *CaNu* to overall LPP variability are relatively balanced. Consequently, LPPs in the two regions are not necessarily correlated within the same specimen or across different specimens. Additionally, ROT increases lipid peroxidation to varying degrees in different individuals in both brain regions.

The HCDA revealed the extent of similarity of LPPs among rats subjected to various experimental treatments (Figure 6b). The analyses show that LPPs exhibit intrinsic inconsistency, both within and between experimental groups. This phenotypic inconsistency is less pronounced in intact rats but increases from that in neuro-restored and neuro-protected rats to those that were administered ROT. The observation that neuro-protected, neuro-restored, and intact rats cluster distinctly from ROT-administered rats suggests that Nano-PSO exerts neuroprotective and neuro-restorative effects. Nevertheless, in all cases, the level of improvement achieved is insufficient to keep LPP values within normal parameters. However, the neuro-protected rats coded as R6 and R7, and, to a lesser extent, R3 and R4 may benefit from continued Nano-PSO supplementation.

### 2.7. Assessing TPs Variance and Similitude

Neurodegenerative processes extend beyond neurological boundaries [46,47,48,49]. Their origin and progression are conditioned by myriad systemic and local metabolic factors that interact throughout an individual’s lifespan [9], shaping the trajectories of the disease from the prodromal stage to diagnosis and beyond [50]. This holds true, even in murine experimental units that model these diseases [51]. Therefore, we conducted a series of molecular and biochemical analyses to further investigate variations in the metabolic phenotype in intact, neuro-protected, neuro-restored, and ROT-administered rats. We begin by studying the expression of antioxidants and metabolic biomarkers.

The PCA reveals the extent of TP variation in the *CaNu* of intact, neuro-protected, neuro-restored, and ROT-administered rats (Figure 7a). *CaNu* TP variability is greatest in intact rats, followed, in decreasing order, by the populations of neuro-restored, neuro-protected, and ROT-administered rats. *CaNu* TP variance is notable within and between groups, indicating their low internal consistency. As a result, a great deal of intergroup overlap is observed. Despite this variability, the relative position, dimension, and orientation of the variance ellipses in the PCA dimensional space supports the theory that Nano-PSO conferred the greatest benefits when used as a neuro-restorative. The magnitude of such a benefit is moderate, individually variable, and not categorical.

A closer examination of the PCA results also reveals that, in neuro-restored rats, the variability of TPs is predominantly influenced by the transcription levels of GPX1, α-syn, GLUT4, and GLUT3. For neuro-protected and ROT-administered rats, the variability of TPs is more profoundly influenced by the transcription levels of CAT, SOD, and TH. Lastly, TP variance in intact rats is driven by the transcription levels of α-syn, GLUT4, GLUT3, SOD, and TH. However, these are generic transcriptional patterns that vary notably across specimens, both within and between the experimental groups of rats. Overall, these results support the theory that Nano-PSO, when used as a neurorestorative, improves *CaNu* TP variability more compared to its use as a neuroprotectant. Despite these heterogeneous and moderate effects, numerous neuro-restored and neuro-protected rats benefited to varying degrees from the consumption of Nano-PSO.

The HCDA evaluated the similarity of *CaNu* TPs among rats from different experimental groups (Figure 7b). The analyses revealed that, regardless of the experimental groups, the majority of *CaNu* TPs exhibit distinct differences from one another; however, TPs cluster non-categorically. While the results suggest that Nano-PSO provides some degree of neuro-restoration and, to a lesser extent, neuroprotection, the overall improvement achieved is insufficient to maintain the TPs within the normal range. However, specimens from the neuro-restored rats coded as R6, R7, and R8 appear to benefit from continued Nano-PSO supplementation, as they cluster close to intact rats. Surprisingly, some intact and ROT-administered rats also cluster near each other, implying that, in certain specimens, rotenone does not significantly deteriorate the transcription of the assessed genes as expected.

### 2.8. Assessing AOPs Variance and Similitude

The onset, progression, and outcome of neurodegenerative processes, regardless of the patient’s phenotype, are influenced by the cumulative and additive effects of systemic and local oxidative stress, which are associated with mitochondrial dysfunctions (e.g., [19]) dopamine metabolism (e.g., [52]), and other sources of oxidative radicals (e.g., environmental pollutants [53]). Therefore, monitoring antioxidative systemic defense is crucial for designing preventive and restorative measures to combat neurodegeneration in susceptible and diagnosed individuals. Punicic acid increases the activity of circulating CAT, GPX1, and SOD in plasma following arsenic exposure (e.g., [54]). Thus, we estimated these biomarkers in the form of AOPs in intact, neuro-protected, neuro-restored, and ROT-treated rats.

The PCA reveals the extent of AOP variation in intact, neuro-protected, neuro-restored, and ROT-administered rats (Figure 8a). This variability is vast; hence, intergroup overlap is notable. The greatest variability is seen in neuro-protected rats, followed, in decreasing order, by the populations of intact, neuro-restored, and ROT-administered rats. Despite this variability, the relative position, dimension, and orientation of the variance ellipses in the PCA dimensional space supports the theory that Nano-PSO conferred variable protection and restoration, ranging from low to high, in many of the supplemented rats. Furthermore, an in-depth review of the PCA results shows that, in neuro-protected and intact rats, the variability of AOPs is predominantly influenced by the plasmatic activity of GPX1 and CAT. SOD plasmatic activity becomes more relevant for explaining part of the AOP variability in neuro-restored and ROT-administered rats. These generic patterns, nonetheless, vary notably across specimens within and between the experimental groups of rats.

The HCDA evaluated the similarity of AOPs among rats from different experimental groups (Figure 8b). As can be observed from the pattern of clustering, AOPs do not form categorical groups based on treatment. There are, however, numerous neuro-protected and neuro-restored rats that branch near intact rats. These results support the theory that Nano-PSO provides some degree of neuroprotection and neuro-restoration; the effect, nonetheless, varies notably across specimens individually. For some specimens, the improvement achieved is sufficient to approximate the normal ranges. Based on AOPs, the neuro-protected rats coded as R2, R4, R6, R7, and R8 and the neuro-restored rats coded as R1, R2, R3, R5, R6, and R7 would benefit from continued Nano-PSO supplementation. This might be explained by the fact that Nano-PSO buffers oxidative radicals, stabilizes the mitochondrial membrane, and activates other anti-oxidative metabolic cascades [55]. Unexpectedly, ROT-administered rats branch near neuro-protected, neuro-restored, and even intact rats, implying that rotenone does not significantly deteriorate the AOPs.

### 2.9. Assessing GTCPs Variance and Similitude

Punicic acid regulates glucose concentration and lipid profile in circulation (e.g., [56]). Therefore, we analyzed the variance of the GTCPs of intact, neuro-protected, neuro-restored, and rotenonized rats using PCA (Figure 9a). We found GTCP variability to be huge. Thus, the intergroup overlap is notable, and each group’s identity is far from categorical. In this regard, the greatest GTCP variability is seen in intact rats, followed, in decreasing order, by neuro-restored, neuro-protected, and ROT-administered rats. Despite this variability, the relative position, dimension, and orientation of the variance ellipses in the PCA dimensional space supports the theory that Nano-PSO conferred variable restoration and protection, ranging from low to high, in many of the supplemented rats. In addition, a detailed analysis of the PCA results shows that the variability of GTCPs in neuro-restored rats is mainly influenced by cholesterol and triglyceride concentrations. GTCP variability in intact rats is influenced similarly by glucose, triglycerides, and cholesterol.

The HCDA evaluated the similarity of GTCPs among rats from different experimental groups (Figure 9b). None of the GTCPs cluster categorically, suggesting they all are within and between experimental groups, intrinsically inconsistent. Despite the extensive overlapping of GTCPs, Nano-PSO provides some degree of neuroprotection and neuro-restoration, the effects being remarkably individually variable. For some specimens, the improvement achieved is sufficient to approximate normal ranges. Based on GTCPs, neuro-protected rats coded as R6 and neuro-restored rats coded as R5, R6, and R8 would benefit from continued Nano-PSO supplementation. Unexpectedly, ROT-administered rats branch near intact, neuro-protected, and/or neuro-restored rats, implying that ROT does not deteriorate markedly the GTCPs.

## 3. Discussion

The search for natural compounds to prevent, slow down, attenuate, or reverse neurodegenerative processes is a constant task in neuropharmacology (e.g., [57,58,59]). Sometimes, this task is disappointing because the benefits documented in preclinical studies become inconsistent when evaluated in clinical trials (e.g., [60,61,62,63,64,65,66,67]). We attribute this discrepancy [65], firstly, to the belief that phenotypic diversity in experimental animals is much smaller than in humans [6,7,8], and, secondly, to the tendency of researchers to interpret preclinical and clinical results under the umbrella of categorical thinking, establishing causal links where there are only correlations [32,68,69,70,71]. By overlooking interindividual phenotypic variation, we forego the possibility of responsibly transferring benefits to those who need them, even if the benefits are not consistent across the entire population being evaluated. But how can we identify those in need? We propose to do this by incorporating the study of the variation and similarity in phenotypes using non-categorical techniques for data analysis. In this study, we evaluated whether analyzing phenotypic variability and similarity in a population of ROT-treated and phenotypically diverse rats, using PCA and HCDA, could help us identify those that might benefit from Nano-PSO consumption, even if the therapeutic response of the population is not consistent. Others have advocated a similar approach to increasing the reproducibility and translation potential of basic research findings to real-world settings [32,72]. To address the technical limitations of conducting a longitudinal study while enabling inter-individual comparisons, we devised a series of phenotypes and profiles as biomarkers to assess the relative deterioration, neuroprotection, or neuro-restoration effect following ROT administration, with or without Nano-PSO supplementation. Each phenotype was defined based on the statistical interactions between the applied treatments and the maximum number of measured parameters for each phenotype. Our findings indicated that with the exception of MoPs in ROT-administered rats and αSPs in intact rats, the phenotypes of MrPs, αSPs, DPs, LPPs, TPs, AOPs, and GTCPs exhibited intrinsic inconsistency (i.e., non-categorical behavior) both within and across experimental rat groups. Nevertheless, it was consistently possible to identify ROT-treated rats that benefited from Nano-PSO supplementation, whether it was used as a neuroprotective or neurorestorative agent. These results substantiate the central hypothesis of our study.

Furthermore, although we were unable to estimate the intrinsic consistency between phenotypes for each specimen, a limitation that undoubtedly constrains the scope of our study, the results suggest that no single parameter can reliably serve as a biomarker for monitoring deterioration or improvement across the population of ROT-administered rats, irrespective of their experimental group. Given the absence of distinct categorical phenotypes, expanding the range of the parameters used to monitor the overall response may prove more effective than relying on a limited set of biomarkers to track the progression of the pathology.

Why is it crucial to better characterize individual responses to Nano-PSO by using non-categorical criteria to identify the most and least responsive specimens within a population? This variability in phenotypes appears to be indicative of healthier states within the population. This variability facilitates the identification of ROT-administered rats that derive the most benefit from Nano-PSO supplementation. Indeed, Nano-PSO provides neuroprotection or neuro-restoration to varying degrees, depending on each individual’s response to the supplementation and the parameters used to monitor their improvement or deterioration.

Given that Nano-PSO has not been reported as producing adverse collateral effects, our data set suggests that it could be used to supplement the diet of precisely selected patients with less favorable phenotypes. This is of the utmost importance because neurodegenerative processes encompass a spectrum of phenotypes [3,20,23,73,74,75,76,77,78,79]. Our approach may, thus, support the development of more precise and responsible personalized preventive or therapeutic strategies, even for patients subjected to epigenetic factors influenced by conventional therapeutic interventions.

What additional benefits arise from characterizing individual responses to nutraceuticals, using non-categorical criteria to identify the most and least responsive subjects within a population? Considering the phenotypic variability in the neurodegeneration landscape and the individuality that characterizes the natural history of these diseases, the non-categorical interpretation of both basic and clinical studies may expedite the identification of new compounds that could benefit the distinct phenotypes of those patients who are already experiencing neurodegeneration and those susceptible to developing it [80,81].

A fundamental observation made in this work is that, except for the phenotypes αSP and AOP, ROT administration tends to decrease phenotype diversity, whereas Nano-PSO, whether used as a neuro-protectant or neuro-restorative agent, reinstates phenotype diversity, as evidenced by the area occupied by the variance ellipses in PCA and by the degree of branching and phenotype inter-mixing in the HCDA. What does this finding mean? In macro ecosystems, major geological processes that threaten life drive mass extinctions. The remodeling of biogeochemical cycles and of the biosphere–geosphere interactions that follow will promote the diversification of the extant taxa. Exposure to heavy metals induces phenotype diversification in lentil plants by promoting mutations; the mutated plants thrive better under difficult conditions [82]. Chemotherapy imposes a “catastrophic” condition on tumor cells. This event promotes the diversification of phenotypes across tumor cell populations that are likely to gain resistance against the aggressive pharmacological agents to which tumors are exposed during treatments [83]. In a sense, a diagnosis of neurodegeneration signals that a serious underlying condition has been jeopardizing the body’s state of homeostasis for decades. This circumstance surely drives the bodies of the afflicted people into a long-lasting state of allostasis. In the search for solutions, the bodies of affected people unleash their variability potential across the population, conditions that might explain the numerous phenotypes now being described. When the treatment takes the stage, following the clinical indications, phenotype diversity may even increase since therapeutic drugs may modify the trajectory of the disease (i.e., phenotypical therapeutic drift) by increasing the system’s intrinsic noise through epigenetic mechanisms [84,85]. In this regard, it is interesting that long-term levodopa treatment uncovers hidden phenotypes of PD [86].

In clinical settings, it may be challenging to accept that increasing phenotype diversity holds adaptive value, as patients suffering from neurodegenerative conditions do not necessarily experience an improved quality of life. However, if these patients reproduce (with Parkinson’s disease now being documented in individuals as young as their 20 s), they might pass on such adaptive solutions to future generations through epigenetic memories. This hypothetical scenario requires thorough investigation. In our experiments, although ROT administration tended to reduce phenotype variation among the rat population, Nano-PSO increased it, whether being employed as a neurorestorative or a neuroprotective agent. Within our framework, ROT might represent a deleterious event for the rat population, while Nano-PSO could introduce variability, thus perturbing disease trajectories and enhancing phenotype diversity. Consequently, some neuroprotected or neurorestored ROT-administered rats might exhibit better outcomes than others, depending on the parameters used to evaluate health conditions and disease progression.

Given these considerations and our data set, we recommend monitoring not only the efficacy of therapeutics in alleviating disease symptoms but also their potential to generate phenotype diversity. By examining the phenotypes along the spectrum, we increase our chances of discovering biologically driven, innovative solutions that could aid in controlling disease occurrence, causation, and progression.

Lastly, we would like to disclose the pending issues of this study; however, these do not undermine the inferences previously discussed.

Undoubtedly, running a longitudinal study would have increased its robustness. Indisputably, assessing the persistence of health effects of ROT and Nano-PSO over a lifetime would better justify the chronic use of Nano-PSO in clinical settings. Undeniably, varying the doses of ROT and Nano-PSO could have more clearly documented the diversity of therapeutic responses among the different phenotypes. Soon, all these issues must be experimentally addressed to design more rational clinical studies.

Moreover, the pharmacokinetics, pharmacodynamics, and pharmaco-(epi)genetics of Nano-PSO (and ROT) must be elucidated to better understand the mechanisms involved in the responses (and their diversity) of the brain and body to both compounds. This information is necessary for developing personalized therapeutic protocols in the coming years and avoiding the indiscriminate use of Nano-PSO as a preventive supplement due to unscrupulous marketing practices.

## 4. Materials and Methods

### 4.1. Animals

Two-month-old, outbred male Wistar rats (250–260 g of body weight) that were born and raised at the Unidad de Modelos Biológicos, Instituto de Investigaciones Biomédicas, Universidad Nacional Autónoma de México, were randomly assigned to four groups. Intact rats were not subjected to any manipulation. Rats administered ROT received daily subcutaneous doses of this toxicant (1 mg/kg/day) [87] for 21 days and were sacrificed 21 days later (ROT). Nano-PSO (GranaGard^TM^, Granalix Bio Technologies, Ltd., Jerusalem, Israel) was administered orally daily (9–10 a.m.) using a syringe, with the intention to provide either neuroprotection (Nano-PSO/ROT) or neuro-restoration (ROT/Nano-PSO). Neuro-protected rats were treated with Nano-PSO for 21 days, followed by ROT administration for 21 days. Neuro-restored rats were administered ROT for 21 days, followed by Nano-PSO treatment for 21 days. The dose of Nano-PSO used ranged between 160 and 300 mg/day, depending on the body weight of each rat. This dosage was estimated, based on the dose recommended in Mexican adults by the supplier (980 mg/day [88]). These doses were 2.96 to 4.62 times higher than those recommended for rats in a previous study [89] (600 mg/kg/day of extract ≈ 180 mg punicalagin/kg/day ≈ 54 mg/day per rat). The LD_50_ estimated in this study was 5000 mg/kg body weight, suggesting that punicic acid is essentially innocuous, even at very high doses [89]. After consumption, punicic acid reaches the plasma via lymphatic drainage, following intestinal absorption [90]. Once in the vascular compartment, it is primarily taken up by the intestine, liver, kidney, adipose tissue, heart, mammary gland, and brain, where it is transformed into conjugated 18:2 9-*cis*, 11-*trans* linoleic acid (CLA), possibly through a Δ13-saturation reaction. Conversion rates in the brain might be lower since CLA accumulation in this organ is minimal [90,91,92]. However, the nano-formulation of punicic acid readily crosses the blood-brain barrier and reaches the brain at therapeutic concentrations after ingestion, at least in mice [29]. For a detailed understanding of the experimental design, please refer to the diagram titled “Experimental design” in the Appendix A. For technical details about the protocol used to produce Nano-PSO, please refer to the PDFs titled Patent 10,154,961 and Patent US20220339101A1 in the Appendix A.

All rats were housed in groups of three individuals per cage, kept under an inverted light cycle (19:00–7:00, lights on; 7:00–19:00, lights off), standard conditions of temperature (22 °C), and relative humidity (50%), and having free access to food and water. The behavioral assessment and biochemical sampling were conducted during the rats’ active phase (lights off). Protocols describing animal handling and experimentation were subjected to evaluation and were approved by the Comité para el Cuidado y Uso de Animales de Laboratorio (ID: 3297).

### 4.2. Motor Coordination Tests

Motor coordination was evaluated using the inclined beam test [93]. Briefly, the Ctrl (n = 10), ROT (n = 10), Nano-PSO/ROT (n = 10), or ROT/Nano-PSO (n = 10) rats were each placed on the lower end of the beam and allowed to climb it until reaching the beam’s upper far end, where a safe place waited for them to be securely sheltered; behavioral testing was performed between 9:00 and 11:00 a.m. The beam (2 m long, 24 mm in width) was kept tilted at an angle of 15°. Each rat was subjected to a reference test to rule out motor dysfunctions before the treatments started. A second test and third test were carried out in all the rats at 21 days and 42 days after starting the treatments. The time taken (in seconds) by each rat to traverse the inclined beam (i.e., the latency) was recorded. The test was considered to have failed if the rat did not reach the top of the beam in less than 120 s; a failed trial was assigned a value of 120 s.

### 4.3. Dopaminergic Neuron Counts

Ctrl (n = 6), ROT (n = 6), Nano-PSO/ROT (n = 6) or ROT/Nano-PSO (n = 6) rats were euthanized with pentobarbital (18.5 mg/kg of bodyweight, i.p., PiSA, Ciudad de México, México) and perfused transcardially with 250 mL of phosphate-buffered saline (PBS; 0.1 M, pH 7.4), followed by 250 mL of buffered paraformaldehyde (4%). At the end of the procedure, the carcasses were decapitated and the brains were removed and then cryopreserved in a buffered solution containing, first, 15% and then 30% sucrose; they remained stored, immersed in this last solution until use. On the day of the experiment, the brains were frozen in dry ice. Cryostat coronal brain slices (40 μm thick) were cut and then collected and incubated in PBS supplemented with 0.3% Triton X-100 for 30 min at room temperature. After a gentle wash in PBS, the sections were incubated with primary polyclonal antibodies raised in rabbits against tyrosine hydroxylase (TH, Cat. No. sc-14007; Santa Cruz Biotechnology, Dallas, TX, USA), diluted 1:1000 in PBS supplemented with 0.3% Triton (PBSt) overnight at 4 °C. After a thorough wash, slices were incubated for 2 h at room temperature with biotin-conjugated, secondary polyclonal antibodies raised in goat anti-rabbit IgGs (Cat. No. BA-1000; Vector Laboratories, Newark, CA, USA), diluted to 1:2000 in PBSt. This step was followed by three washes in PBS. Then, the slices were incubated for 2 h at room temperature with avidin-peroxidase, prepared as recommended by the supplier (Standard Elite kit, Cat. No. PK-6100; Vector Laboratories). After two washes in PBS, the sections were finally incubated with a solution containing 2,2-diaminobenzidine and hydrogen peroxide, as recommended by the supplier (peroxidase substrate kit DAB (Cat. No. SK-4100; Vector Laboratories). The stained slices were mounted onto gelatin-subbed slides and cover-slipped with Cytoseal 60 (Cat. No. Cat. 8310-4; Thermo Fisher Scientific, Ciudad de México, Mexico). The histological material thus obtained was used to manually count TH-positive neurons (20×) in a volume fraction of the *Substantia Nigra pars compacta* (*SNpc*) located between −5 and −6 mm anteroposteriorly from the bregma. Six consecutive sections per rat were used to obtain these estimates. The value of the volume fraction sampled was calculated by multiplying the area of *SNpc* (estimated after delineating its boundaries; ImageJ software, version 1.53t, NIH, Bethesda, ML, USA) by the slice thickness. For the study, we only counted TH+ neurons showing central, unstained nuclei. This decision reduced the possibility of having double cell counts in consecutive sections.

### 4.4. Concentrations of Dopamine (DA), 3,4-Dihydroxyphenylacetic Acid (DOPAC), and Serotonin (5-HT) by Reverse Phase, High-Performance Lipid Chromatography

Ctrl (n = 8), ROT (n = 8), Nano-PSO/ROT (n = 8), or ROT/Nano-PSO (n = 8) rats were euthanized and decapitated. The brains were rapidly removed from the skull and hemisected. Samples of the caudate nucleus (*CaNu*) and *SNpc* were dissected under microscopic view, collected in centrifuge tubes, frozen in dry ice, and stored at −75 °C until used. Neurotransmitters were isolated as previously described [94,95,96,97].

Tissue samples (10–100 mg of tissue), contained in a microtube, were thawed to extract the neurotransmitters; dopamine (DA), 3,4-dihydroxyphenylacetic acid (DOPAC), and serotonin (5-HT) were extracted in a 1.5 mL microfuge tube by adding 400 μL of a buffer composed of 5% ascorbic acid, 200 mM sodium phosphate, 2.5 mM L-cysteine, and 2.5 mM EDTA. Three 20-s ultrasonic pulses (30% amplitude) were used to homogenize each sample. Then, the protein fraction was precipitated by adding 100 μL of 0.4 M perchloric acid while keeping the solution at 20 °C for 20 min. Supernatants containing DA, DOPAC, and 5-HT were collected after centrifugation at 12,000 rpm for 20 min at 4 °C. These samples were concentrated by first passing them through a 0.22 pore size filter and then through a solid phase extraction column (Strata C18-E, 55 μm, 70 A, 100 mg^−1^ mL; Phenomenex, Torrance, CA, USA). The extraction column was activated by passing 250 μL of mobile phase B buffer (FMB; 0.1% trifluoroacetic acid in acetonitrile) through it, then it was equilibrated by flowing 250 μL of mobile phase A (FMA; trifluoroacetic acid 0.1% diluted in deionized water). Finally, each of the collected samples was loaded onto the column and the neurotransmitters contained therein were eluted using 250 μL of FMA: FMB (25:75 *v*/*v*) passed through the extraction column.

DA, DOPAC, and 5-HT concentrations were determined by using reversed-phase high-performance liquid chromatography. Our system was integrated by an APU-2089 plus pump (Jasco Inc., Easton, MD, USA), an AS-2057 plus autosampler (Jasco), and an X-LC™3120FP fluorescence detector (Jasco). The instruments were controlled by the ChromNav software, version 2.4 (Jasco). Chromatographic runs were performed using a CAPCELL PACK MGII C18 column (300 Å, 5 μ, 4.6 × 250 mm, SHISEIDO, Tokyo, Japan) at 30 °C. The column was equilibrated by flowing FMA for 120 min at a rate of 0.8 mL/min. The samples (10–50 μL) were then injected into the HPLC column. DA, DOPAC, and 5-HT were separated, using a gradient created by flowing first FMA (3 min/0.8 mL/min) and then FMB. The first FMB ramp went from 0–10% between minutes 3 and 33, and the second ramp went from 10 to 30% between minutes 33 and 39; for each ramp, the flow rate was kept at 0.8 mL/min. This maneuver removed the non-targeted metabolites. The plateau of 30% FMB was maintained for 4 min. Thereafter, FMA was perfused from minute 45 up to minute 60 (flow rate 0.8 mL/min), in preparation for the next injection. This protocol was repeated for each sample.

DA, DOPAC, and 5-HT were detected through fluorometry (gaining set at 100 and attenuation factor at 32). They were all excited at 280 λ and their emission was identified at 315 λ. DA, 5-HT, and DOPAC showed retention times of 13.6, 26.2, and 29.2 (±5%) minutes, respectively. Data were acquired and processed using ChromNav software (Jasco). Analyte concentrations are expressed as pmol/mg tissue; all samples were analyzed in duplicate. Our chromatographic method was validated prior to sample analysis; it had a precision and recovery level of ≥95%. The calibration curve showed an R^2^ ≥ 0.99 for the three analytes. Calibration curves were performed in a linear range, as follows: DA from 0.625 to 100 pmol; DOPAC from 12.5 to 1000 pmol; and 5-HT from 0.125 to 20 pmol.

### 4.5. Concentration of α-Synuclein by ELISA

Ctrl (n = 8), ROT (n = 8), Nano-PSO/ROT (n = 8), or ROT/Nano-PSO (n = 8) rats were euthanized with pentobarbital and decapitated. The brains were removed, placed on a prechilled plate, and hemisected. Once exposed, samples of the *CaNu* and *SNpc* were dissected under microscopic guidance, rinsed in PBS (0.02 M; pH 7.2), and weighed. The samples (25 mg) were sonicated (20 Hz; 3 times/1 min each at 4 °C) in 100 µL of PBS (0.02 M; pH 7.4), supplemented with the complete protease inhibitor cocktail (Cat. No. 4693124001, Roche, Ciudad de México, Mexico). The homogenates were then centrifuged at 12,000 rpm for 5 min at 4 °C, and the supernatants were stored at −80 °C until use. Protein determination was conducted using 10 µL aliquots with the bicinchoninic acid protein quantification assay kit (Cat. No. 23225; Pierce Thermo Fisher Scientific, Ciudad de México, Mexico). The concentration of α-synuclein was estimated in both *CaNu* and *SNpc* supernatants using the rat α-synuclein ELISA kit, following the protocol provided by the supplier (Cat. No. LS-F23251, LSBio, Shirley, MA, USA).

### 4.6. Estimation of Lipid Peroxidation by Colorimetry

Ctrl (n = 8), ROT (n = 8), Nano-PSO/ROT (n = 8), or ROT/Nano-PSO (n = 8) rats were euthanized with pentobarbital and decapitated. Their brains were rapidly removed, placed on a prechilled plate, and hemisected. Then, the *CaNu* and *SNpc* samples were dissected under microscopic guidance, rinsed in PBS (0.02 M; pH 7.2), weighed, put into a centrifuge tube, and sonicated (25 mg) in radioimmunoprecipitation (RIPA) buffer (25 mM Tris–HCl pH 7.4, 150 mM NaCl, 1% NP-40, 0.25% Na-deoxycholate, 1 mM and SDS 0.1%) supplemented with the complete protease inhibitor cocktail (Cat. No. 4693124001, Roche) for 15 s at 40 Hz while ice-cold. The samples were then centrifuged at 1600× *g* for 10 min at 4 °C. The supernatants were collected and stored at −80 °C until use. Protein determination was performed in 20 µL supernatant samples, using the bicinchoninic acid protein quantification assay kit (Cat. No. 23225; Pierce Thermo Fisher Scientific). Lipid peroxidation was estimated by determining malonyl dialdehyde levels (MDA; μM/mg of protein) using the Thiobarbituric Acid Reactive Substances (TBARS) kit, following the protocol recommended by the supplier precisely (Cat. No. KA1381, Abnova, Taipei, Taiwan [98,99]).

### 4.7. Estimation of the Activity of Oxidative Enzymes by Colorimetry

Ctrl (n = 8), ROT (n = 8), Nano-PSO/ROT (n = 8), or ROT/Nano-PSO (n = 8) rats were deeply anesthetized with pentobarbital. Each rat was placed on its back and the thoracic cavity was opened. The heart’s left ventricle was punctured, and blood samples were withdrawn (no heparin was used) using a 5 mL syringe armed with a 23 GI needle. After disengaging the needle, the syringe content was emptied into 2.5 mL Eppendorf tubes. The samples thus collected were allowed to coagulate at room temperature. These tubes were centrifuged at 15,000 rpm for 10 min at 4 °C, and the supernatants were collected in aliquots (200 μL) that were stored at −80 °C until use. In the meantime, each carcass was decapitated. Their brains were rapidly removed, placed on a pre-chilled plate, and hemisected. The *CaNu* and *SNpc* samples were dissected bilaterally under microscopic guidance, weighed, and collected into 2.5 mL Eppendorf tubes placed in ice. These samples were later used to extract total RNA (see below).

Serum catalase activity (CAT; Catalog No. ab83464 [100], superoxide dismutase (SOD; Catalog No. ab65354 [101]), and glutathione peroxidase (GPx; Catalog No. ab102530 [102]) kits were used, following the recommendations provided by the supplier (Abcam, Cambridge, UK).

### 4.8. Gene Expression Profiling

Ctrl (n = 8), ROT (n = 8), Nano-PSO/ROT (n = 8), and ROT/Nano-PSO (n = 8) rats were euthanized with pentobarbital and decapitated. Their brains were removed, placed on a prechilled plate, and hemisected. Samples of *CaNu* were dissected under microscopic guidance and were used to isolate total RNA, following the protocol suggested by the manufacturer (TRIzol^®^Reagent, Life Technologies, Thermo Fisher Scientific, Waltham, MA, USA). Briefly, tissue samples were homogenized in TRIzol (1 mL), vortexed, and centrifuged with the Velocity 18R Pro centrifuge (Dynamica Techcomp Company, Kirkton Campus, Livingston, UK) at 12,000 rpm for 10 min at 4 °C. The supernatants were collected and separated in phases after adding 200 μL of chloroform (Sigma-Aldrich^®^ St. Louis, MO, USA) for 10 min at room temperature. During phase separation, the samples were vortexed once every three minutes. Thereafter, the samples were again centrifuged at 10,200 rpm for 10 min at room temperature. The aqueous phase was recovered and mixed with 500 μL of isopropanol (Sigma-Aldrich^®^ St. Louis, MO, USA). This solution was vortexed once every three minutes while being kept at room temperature for 10 min, after which, the samples were centrifuged at 14,000 rpm for 10 min at 4 °C. The supernatants were removed and discarded. The pellets of RNA were washed by vortexing with 75% ethanol (Sigma-Aldrich^®^ St. Louis, MO, USA). Centrifugation followed at 14,000 rpm for 10 min at 4 °C. The total RNA was finally solubilized in DEPC-treated water (Thermo Fisher Scientific Inc., Waltham, MA, USA), after removing the ethanol and being dried with the aid of a Thermo Scientific Savant SpeedVac Concentrator (Thermo Fisher Scientific Inc., Waltham, MA, USA). RNA integrity was verified by electrophoresis with the Thermo Scientific Owl^®^ EC-300XL Compact Electrophoresis Power Supply System Concentrator (Thermo Fisher Scientific Inc., Waltham, MA, USA) through 1.5% agarose (Sigma-Aldrich^®^, St. Louis, MO, USA) gels labeled with ethidium bromide (Sigma-Aldrich^®^, St. Louis, MO, USA) for 20 min at 65 V. Its purity was established, based on the 260/280 ratio (~1.8), itself derived from the Qubit^®^ 3.0 Fluorometer (Thermo Fisher Scientific Inc., Waltham, MA, USA) readings. The total RNA samples were finally stored at −80 °C. cDNA synthesis was performed in the thermal cycler Techne^®^
^3^Prime (Cole-Parmer Ltd., Beacon Road, Stone, UK), using the SensiFAST cDNA synthesis kit (Bioline Reagents Ltd., Humber Road, London, UK), according to the manufacturer’s instructions. Shortly afterward, we prepared a master mix for each sample, containing the total RNA (4 μL), 5× TransAmp buffer (4 μL), reverse transcriptase (1 μL), and Dnase/Rnase free water (11 μL). Reverse transcription was carried out using the following parameters. Step 1: primer annealing, 25 °C/10 min. Step 2: Reverse transcription, 42 °C/15 min. Step 3: Inactivation, 85 °C/5 min. Step 4: Hold at 4 °C. The relative expression of each gene being evaluated was determined based on the probes annotated in the rat’s transcriptome library (Rat Universal Probe Library). Sense and antisense and specific oligonucleotide primers were designed (Table 1) with the aid of the Probe Finder software version 2.45, based on the gene sequences reported in the universal probe library (https://lifescience.roche.com/global/en/home.html, (accessed on 18 February 2022). RT-PCR was conducted using TaqMan-type reaction mixtures (SensiFAST Probe No-ROX kit; Bioline Reagents Ltd., Humber Road, London, UK) with the aid of a RT-qPCR thermal cycler, Techne^®^ Prime Pro 48 (Cole-Parmer Ltd., Beacon Road, Stone, UK), as recommended by the supplier. The protocol used was as follows. Step 1: Initial denaturation conducted at 95 °C for 10 min. Step 2: 45 cycles of amplification conducted each at 94 °C for 10 s, 60 °C for 20 s, and 72 °C for 5 s. The conditions of the amplification were defined based on the comparative parameter threshold cycle method [103]. The 18S rRNA amplification values were used as internal controls, and the entire data set thus obtained was analyzed with the Pro Study Software Techn, version 5.2.17 (Cole-Parmer Ltd., Beacon Road, Stone, UK).

### 4.9. Blood Glucose and Lipid Estimates

Glucose (mmol/dL; Accutrend glucose strips, ref. 11447475), cholesterol (mg/mmol/dL; Accutrend cholesterol strips, ref. 11418262), and triglycerides (mmol/dL; Accutrend triglycerides strips, ref. 11538144) concentrations were estimated in blood samples withdrawn from the tail. Measures were obtained using an automated strip reader system, the Accutrend Plus monitor (ref. ROC-FL 505050472023, Roche Diagnostics International, Rotkreuz, Switzerland), validated by Bland and Blad in 1986 [104].

### 4.10. Statistics

The statistical procedures and analyses presented in this work were conducted on a data set that was first organized in hierarchical contingency tables (Excel, Microsoft, Redmon, WA, USA), and then turned into matrixial arrangements (Excel, Microsoft). This allowed the data set to be imported to an R environment (see the Appendix A: Statistics Technical Report; 12 New Data Base 15.02.23_B-E.xlsl; g-ANOVA.R; g-data entry.R; g-PCA.R, g-tree.R; Nano-PSO-v12.Rmd; nPSO-V12-comparison.Rmd; nPSO-V12-dendrograms.Rmd; nPSO-V12-pca.Rmd; 3.Analisis.Rproj). After importing the data set, this was transformed into the appropriate structure by which to carry out the following statistical analyses: (a) descriptive statistics in which the null hypothesis was evaluated by the comparisons conducted, among experimental groups, for each parameter considered in the study (one-way, non-parametric Kruskal–Wallis tests for unpaired data, followed by post hoc tests; *p* significance > 0.05); (b) principal component analysis; (c) dendrograms; (d) outlier identification (see the Appendix A).

## 5. Conclusions

This study shows that non-categorical methods of data analysis, in particular PCA and HCDA, can be used to identify and select ROT-administered rats that could benefit from the continuous consumption of Nano-PSO, even when the neuroprotective or neuro-restoring response is far from categorical across the population, irrespective of the biomarker used to monitor disease progression. Based upon these observations, we believe that the translational success of basic science studies investigating Nano-PSO (and other possible nutraceuticals and pharmaceuticals) as neuro-protectant and/or neurorestorative agents for neurodegeneration could be enhanced by incorporating non-categorical methods of data analysis into the clinical arena. By employing these methods for proper patient screening and selection [105], patients experiencing neurodegenerative conditions might benefit from Nano-PSO (and other nutraceuticals), even if the overall therapeutic outcomes for the entire population are intrinsically inconsistent. It is worth emphasizing that during the process of personalizing treatments, special care must be taken not to exaggerate the pharmaceutical properties and scope of the nutraceutical, especially if it is used as a neuroprotectant. In addition, it is essential to publicly inform physicians about the need to strictly select patients or susceptible individuals (e.g., those living in highly polluted areas, those with occupational exposure to insecticides, pesticides, and herbicides, or those with a family history of neurodegenerative diseases) to avoid abuses by unscrupulous sellers and dishonest commercial practices. This approach will enable physicians to make rational decisions that help patients and help the patients’ families to avoid financial strain.

## Figures and Tables

**Figure 1 ijms-25-12635-f001:**
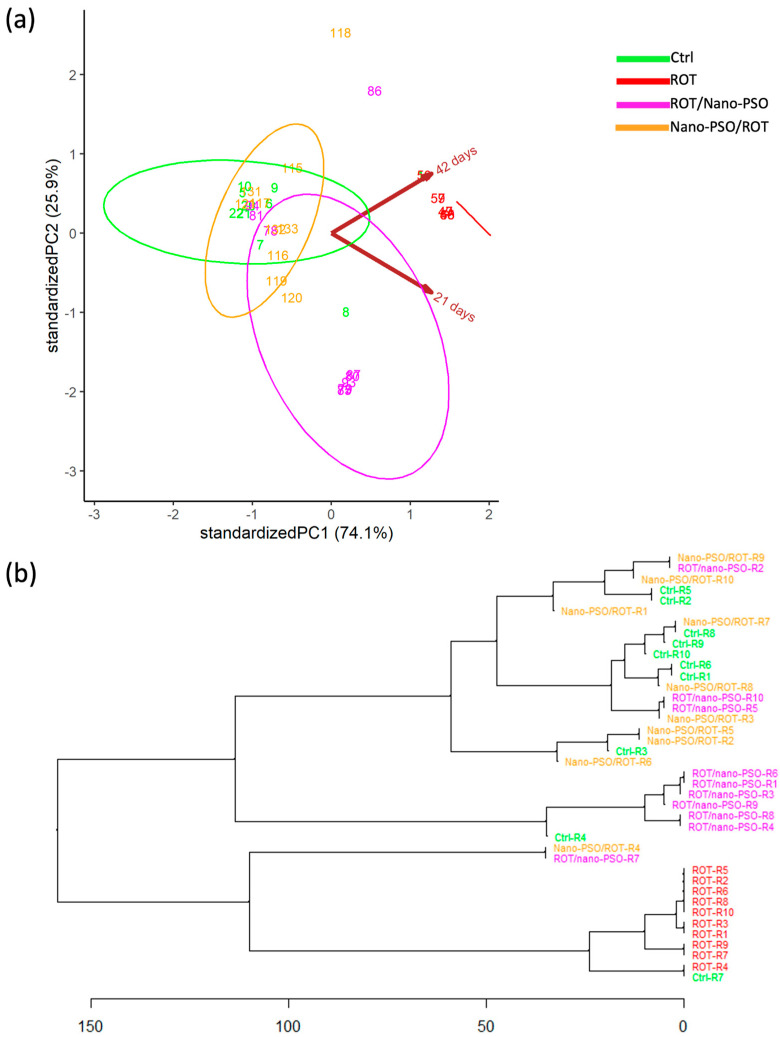
Motor phenotypes (MoPs). (**a**) Principal component analysis plotting the MoPs of intact (Ctrl), neuro-protected (Nano-PSO/ROT), neuro-restored (ROT/Nano-PSO), and rotenonized (ROT) rats. The graph shows the magnitude of MoP variability and the probability of their variance (ellipses). (**b**) Complete linkage clustering dendrogram that shows the degree of similitude of MoPs within and between Ctrl, Nano-PSO/ROT, ROT/Nano-PSO, and ROT rats. The greatest phenotypic similarity is found when the phenotypes are adjacent, are close to the branching points, and appear along with less complex branching patterns.

**Figure 2 ijms-25-12635-f002:**
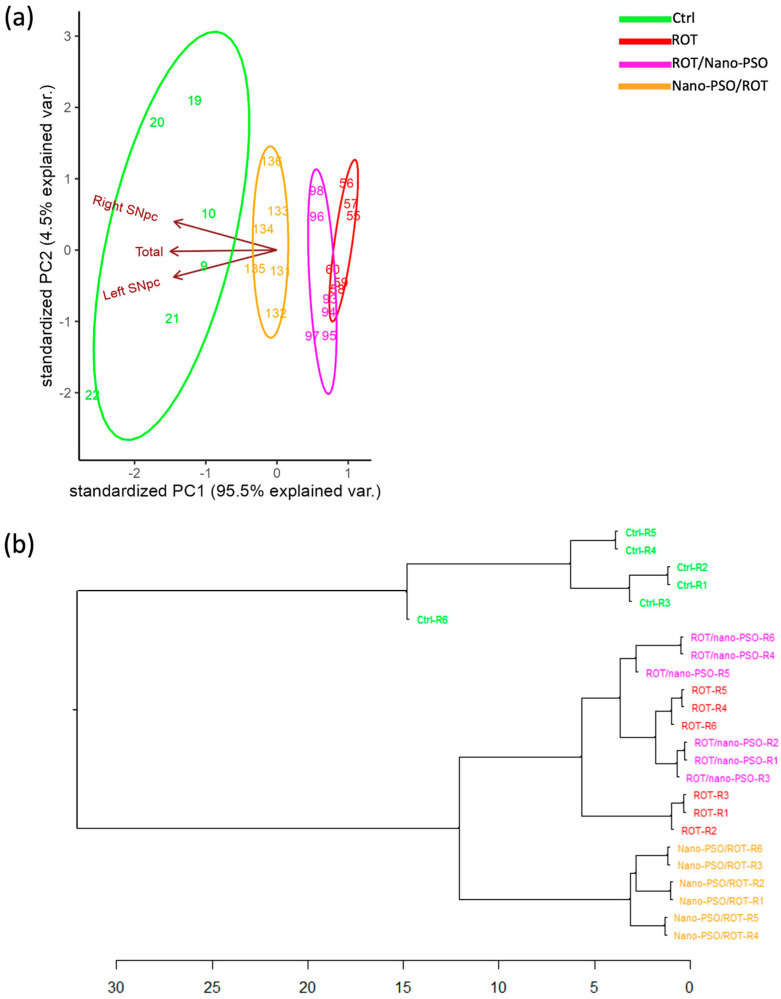
Morphotypes (MrPs). (**a**) Principal component analysis that plots the MrPs of Ctrl, Nano-PSO/ROT, ROT/Nano-PSO, and ROT rats. The graph shows the magnitude of MrP variability and the probability of their variance (ellipses). (**b**) Complete linkage clustering dendrogram that shows the degree of similitude of MrPs within and between intact Ctrl, Nano-PSO/ROT, ROT/Nano-PSO, and ROT rats. The greatest phenotypic similarity is found when the phenotypes are adjacent, are close to the branching points, and appear along with less complex branching patterns. Abbreviations: *Substantia nigra pars compacta* (SNpc).

**Figure 3 ijms-25-12635-f003:**
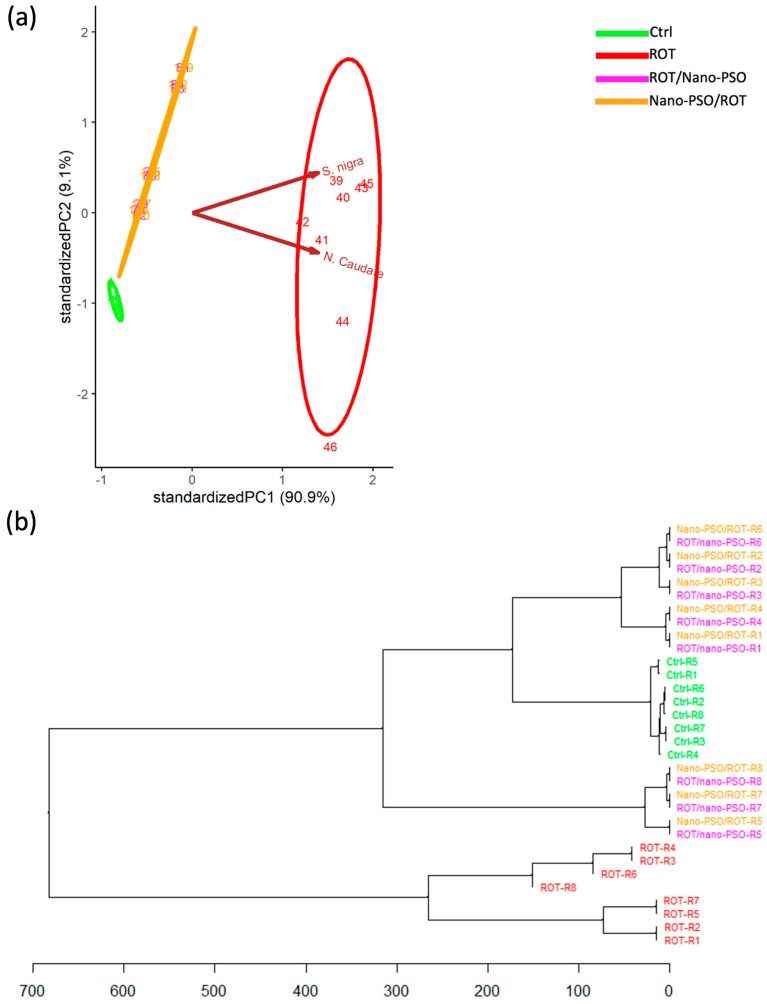
α-Synuclein phenotypes (αSPs). (**a**) Principal component analysis that plots the αSPs of Ctrl, Nano-PSO/ROT, neuro-restored, ROT/Nano-PSO, and ROT rats. The graph shows the magnitude of αSP variability and the probability of their variance (ellipses). (**b**) Complete linkage clustering dendrogram that shows the degree of similitude of αSPs within and between Ctrl, Nano-PSO/ROT, ROT/Nano-PSO, and ROT rats. The greatest phenotypic similarity is found when the phenotypes are adjacent, are close to the branching points, and appear along with less complex branching patterns. Abbreviations: *Substantia nigra pars compacta* (S. nigra); Caudate nucleus (N. Caudate).

**Figure 4 ijms-25-12635-f004:**
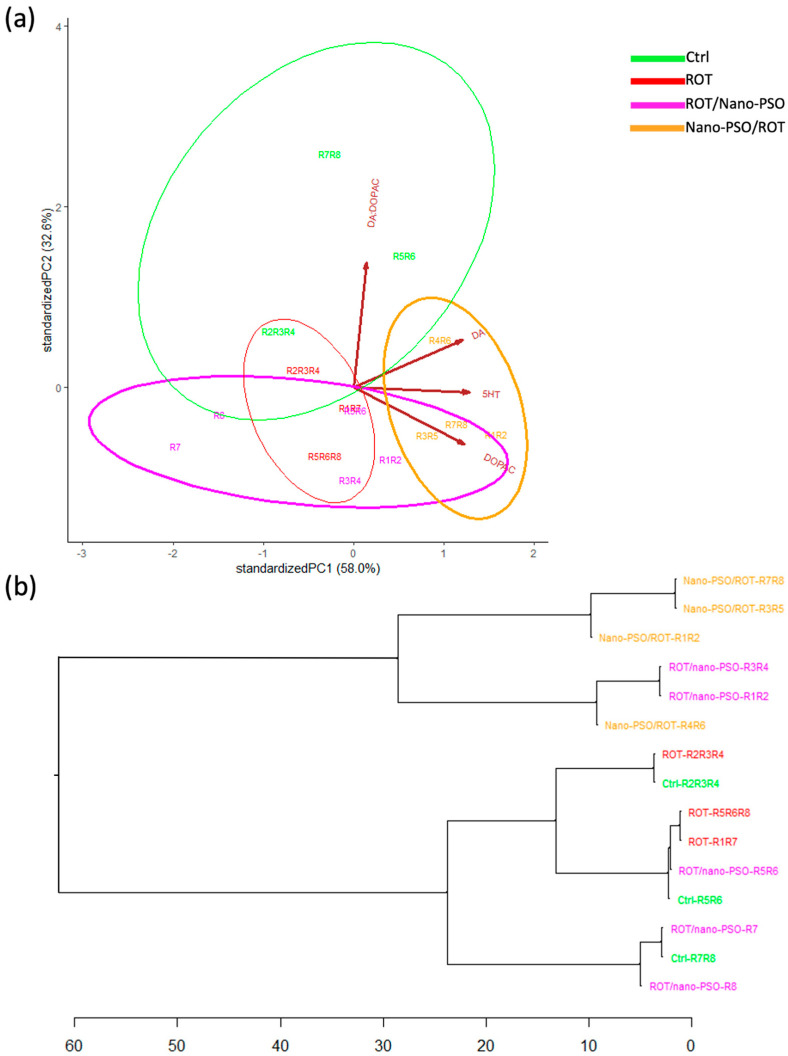
Dopaminergic phenotype (DP) in the *Substantia nigra pars compacta* (*SNpc*). (**a**) Principal component analysis that plots the DP in the *SNpc* of Ctrl, Nano-PSO/ROT, ROT/Nano-PSO, and ROT rats. The graph shows the magnitude of *SNpc* DP variability and the probability of their variance (ellipses). (**b**) Complete linkage clustering dendrogram that shows the degree of similitude of *SNpc* DPs within and between Ctrl, Nano-PSO/ROT, ROT/Nano-PSO, and ROT rats. The greatest phenotypic similarity is found when the phenotypes are adjacent, are close to the branching points, and appear along with less complex branching patterns. Abbreviations: dopamine (DA), 3,4-dihydroxyphenylacetic acid (DOPAC), DA:DOPAC ratio (DA:DOPA), and serotonin (5HT).

**Figure 5 ijms-25-12635-f005:**
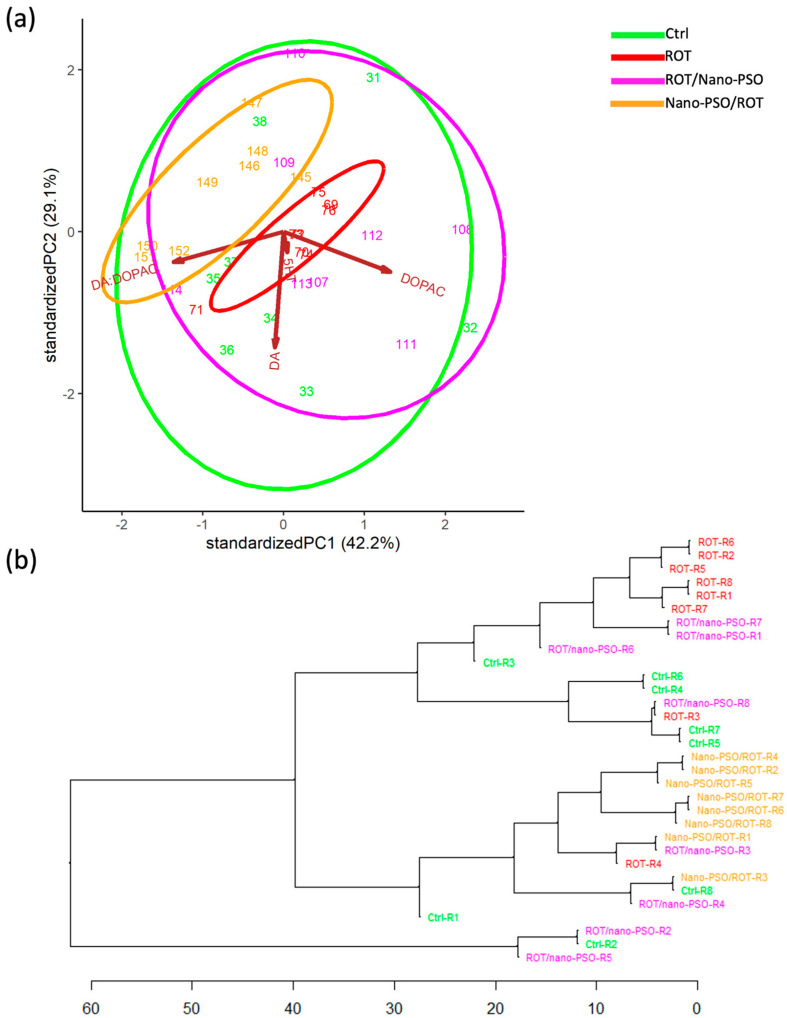
Dopaminergic phenotypes (DP) in the caudate nucleus (*CaNu*). (**a**) Principal component analysis that plots DPs in the *CaNu* of Ctrl, Nano-PSO/ROT, ROT/Nano-PSO, and ROT rats. The graph shows the magnitude of *CaNu* DP variability and the probability of their variance (ellipses). (**b**) Complete linkage clustering dendrogram that shows the degree of similitude of *CaNu* DPs within and between Ctrl, Nano-PSO/ROT, ROT/Nano-PSO, and ROT rats. The greatest phenotypic similarity is found when the phenotypes are adjacent, are close to the branching points, and appear along with less complex branching patterns. Abbreviations: dopamine (DA), 3,4-dihydroxyphenylacetic acid (DOPAC), DA:DOPAC ratio (DA:DOPA), and serotonin (5HT).

**Figure 6 ijms-25-12635-f006:**
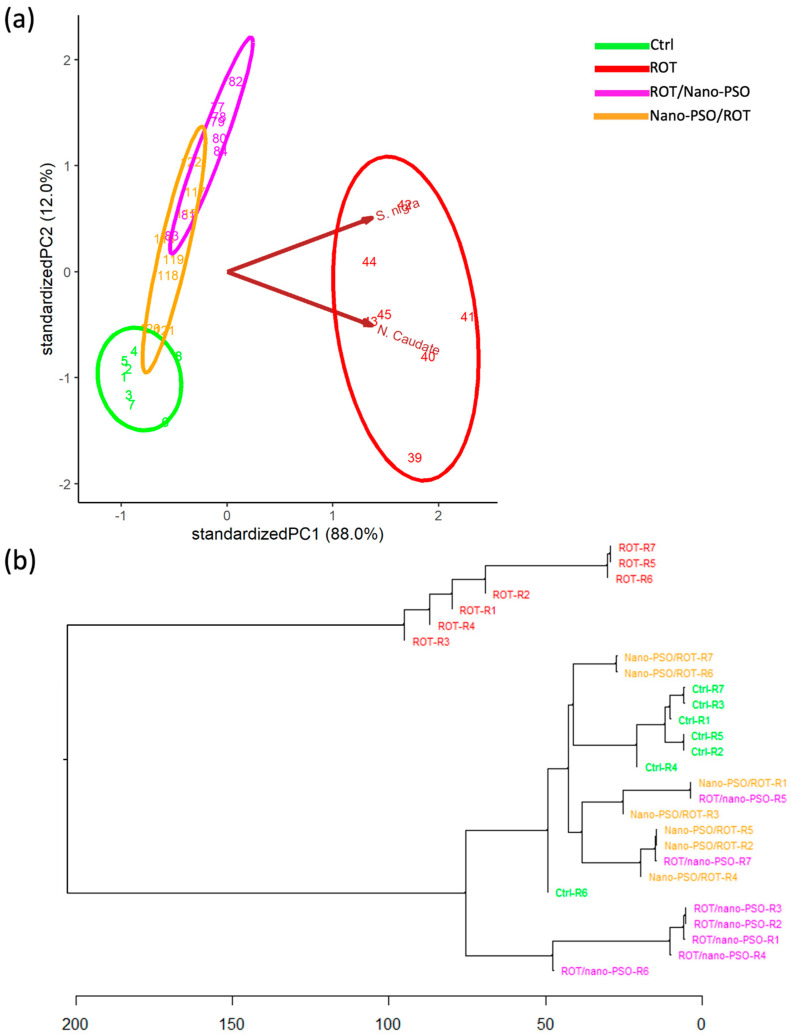
Lipid peroxidation phenotype (LPP). (**a**) Principal component analysis that plots the LPPs Ctrl, Nano-PSO/ROT, ROT/Nano-PSO, and ROT) rats. The graph shows the magnitude of LPP variability and the probability of their variance (ellipses). (**b**) Complete linkage clustering dendrogram that shows the degree of similitude of the LPPs within and between Ctrl, Nano-PSO/ROT), ROT/Nano-PSO), and ROT rats. The greatest phenotypic similarity is found when the phenotypes are adjacent, are close to the branching points, and appear along with less complex branching patterns. Abbreviations: *Substantia nigra pars compacta catalase* (S. nigra); Caudate nucleus (N. Caudate).

**Figure 7 ijms-25-12635-f007:**
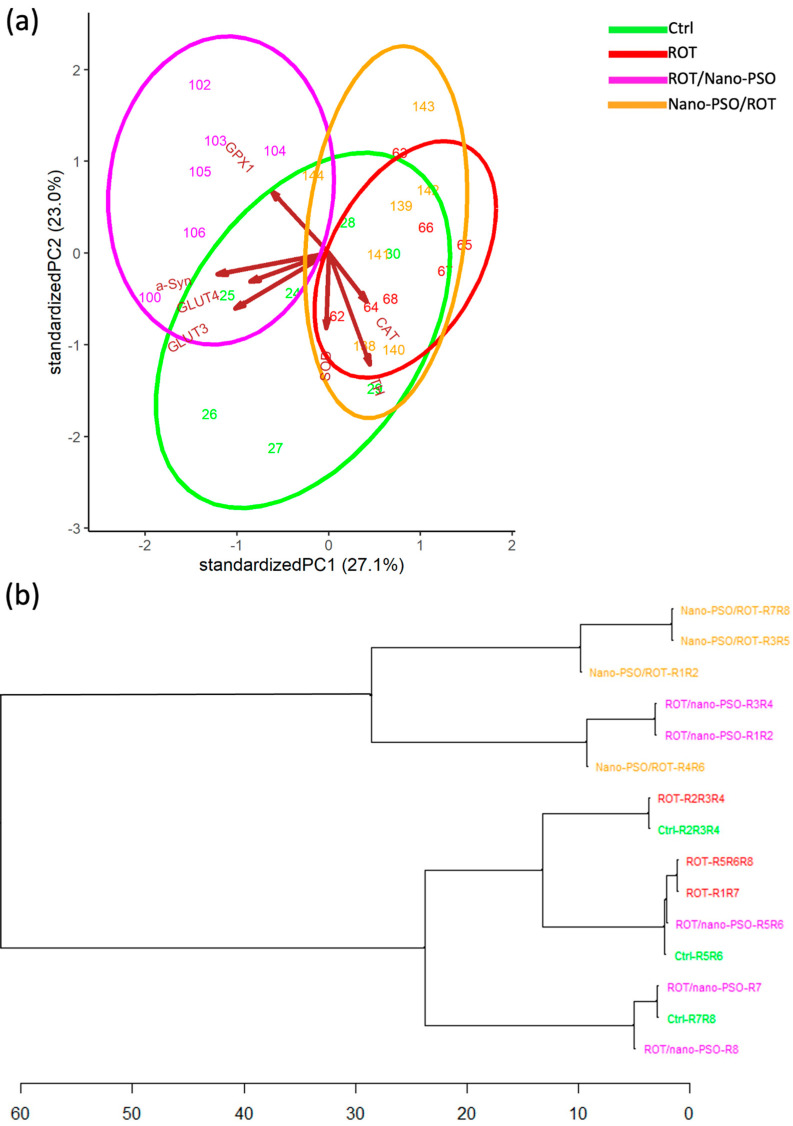
Transcriptomic phenotype (TP). (**a**) Principal component analysis that plots the TPs of Ctrl, Nano-PSO/ROT, ROT/Nano-PSO, and ROT rats. The graph shows the magnitude of TP variability and the probability of their variance (ellipses). (**b**) Complete linkage clustering dendrogram that shows the degree of similitude of *CaNu* TPs within and between Ctrl, Nano-PSO/ROT, ROT/Nano-PSO), and ROT rats. The greatest phenotypic similarity is found when the phenotypes are adjacent, are close to the branching points, and appear along with less complex branching patterns. Abbreviations: catalase (CAT), glutathione peroxidase 1, (GPx1), superoxide dismutase (SOD), glucose transporter 3 (GLUT3), glucose transporter 4 (GLUT4), tyrosine hydroxylase (TH), and α-synuclein (SNCA).

**Figure 8 ijms-25-12635-f008:**
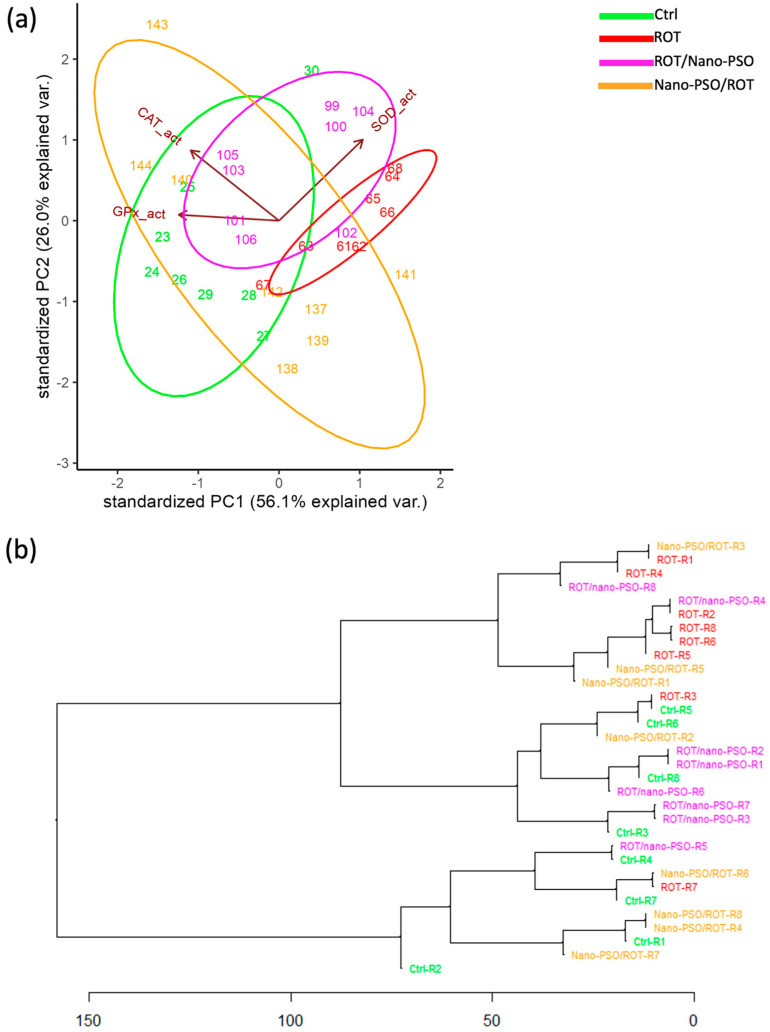
Anti-oxidative phenotype (AOP). (**a**) Principal component analysis that plots the AOPs of Ctrl, Nano-PSO/ROT, ROT/Nano-PSO, and ROT rats. The graph shows the magnitude of the diversity of AOPs and the probability of their variance (ellipses). (**b**) Complete linkage clustering dendrogram that shows the degree of similitude of AOPs within and between Ctrl, Nano-PSO/ROT, ROT/Nano-PSO, and ROT rats. The greatest phenotypic similarity is found when the phenotypes are adjacent, are close to the branching points, and appear along with less complex branching patterns. Abbreviations: catalase (CAT_act), glutathione peroxidase 1 (GPx_act), and superoxide dismutase (SOD_act).

**Figure 9 ijms-25-12635-f009:**
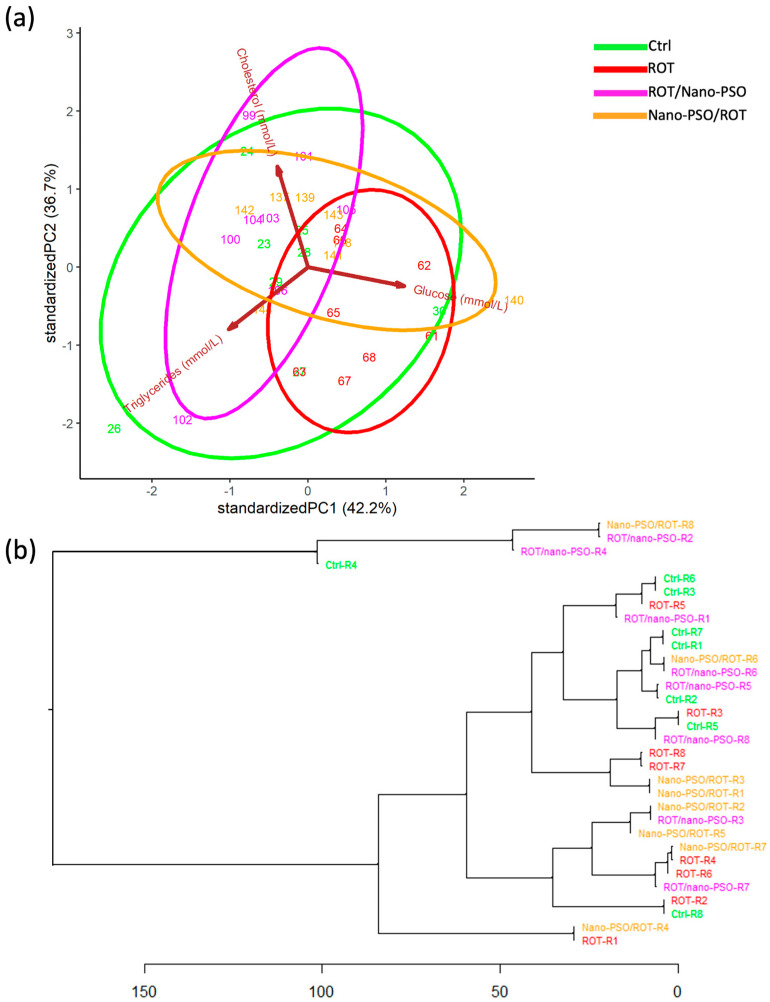
Glucose-triglycerides-cholesterol profile (GTCP). (**a**) Principal component analysis that plots the GTCPs of Ctrl, Nano-PSO/ROT, ROT/Nano-PSO, and ROT rats. The graph shows the magnitude of the diversity of GTCPs and the probability of their variance (ellipses). (**b**) Complete linkage clustering dendrogram that shows the degree of similitude of GTC-profiles within and between Ctrl, Nano-PSO/ROT, ROT/Nano-PSO, and ROT rats. The greatest phenotypic similarity is found when the phenotypes are adjacent, are close to the branching points, and appear along with less complex branching patterns. Abbreviations: cholesterol (EROL); triglycerides (GLYCERIDES).

**Table 1 ijms-25-12635-t001:** Oligonucleotide primers.

Gene	Forward	Reverse
*CAT*	TCACATCTGCAGAGCACTGG	ACTACCCCAACAGCTTCAGC
*GPX1*	GGAATGCCTTAGGGGTTGCT	GCTTTCGCACCATCGACATC
*SNCA*	GACAAAACCAGTGGCAGCAG	CAGTGGTGACTGGTGTGACA
*SOD1*	ATTGGCCACACCGTCCTTTC	GTCCAGCGGATGAAGAGAGG
*TH*	TCCTTCAAGAAGCGGGACAC	TTCTGGAACGGTACTGTGGC
*18S*	CTCAACACGGGAAACTCA	CGCTCCACCAACTAAGAACG
*GLUT3*	CGGAGAAGATGGCCACACAT	TGGAAGAGCGGTTGGAAGAC
*GLUT4*	CAGCGAGGCAAGGCTAGATT	TAGACTCTGGGTGAAGGGGG

Oligonucleotide primers, both sense and antisense.

## Data Availability

Data are contained within the article.

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
