# Peer review of "Non-Categorical Analyses Identify Rotenone-Induced ‘Parkinsonian’ Rats Benefiting from Nano-Emulsified Punicic Acid (Nano-PSO) in a Phenotypically Diverse Population: Implications for Translational Neurodegenerative Therapies"

_ijms, 2024, doi:10.3390/ijms252312635_

Round 1

Reviewer 1 Report

Comments and Suggestions for Authors

The manuscript „The benefit of consuming Nano emulsified punicic acid (Nano-PSO) varies across a population of rotenone-induced parkinsonian albino rats: implications for translational therapies for Parkinson's disease” presents research aimed at developing supporting therapy for Parkinson's disease by application of punicic acid in the form of nanoemulsion. The study utilizes rat model, with investigation how rotenone is affecting the neural functions and the Nano-PSO protecting or restoring them. An important idea in this paper is confronting categorical thinking and phenotypic variation in reaction to treatment (in this case, nutriceutical), with conclusions directed at more personal selection of medicine and supplements.

As the Authors state in the abstract, their results revealed that

1)       the toxicity of rotenone or the benefits of Nano-PSO, the latter used as a neuroprotectant or neurorestorative, varied among rotenonized rats in ranges  that went from specimens that showed substantial damage or benefits to those that were marginally  affected or benefited.

2)       they did not identify any motor, morphological or metabolic parameter or combination of them, which could be used as a reliable index to monitor the deterioration or improvement of the neurometabolic state in the group of rotenonized rats studied.

Several times in the results the Authors describe potential reaction to the test results according to the classical drug development approach “Thus, ruling out the use of a Nano-PSO in phase I clinical studies because the therapeutic response is not significant for the population evaluated, is a mistake, since it reduces the possibilities of many patients of obtaining the benefits of it”. In another fragment, “From a clinical point of view, as a thought experiment, assuming this is a phase one clinical trial, the most receptive subjects could be chosen to continue supplementing Nano-PSO, even if the therapeutic response of the population was poor and would not reach values of significance if parametric methods are used to analyze the data. In our case, nonetheless, numerous neuro-protected rats obtained a great benefit. This was not the case for the neuro-restored rats.”

The suggested benefit for certain phenotypes agrees with postulates of precision medicine and personalized medicine, as the Authors state at the end of Introduction.

The idea of personalized medicine is gaining importance with the progress in studying molecular mechanisms of drug action and subtle differences between individuals. In the manuscript the Authors examined several parameters to assess the effects of rotenone and Nano-PSO, and used hierarchical clustering dendrogram analysis (HCDA) to investigate the phenotyping similarities.

However, the results presented in the manuscript show limited effects of Nano-PSO and irregular reactions and relations in investigated groups. It is believed that animal studies offer more controlled conditions, starting from genetic make-up of animals, and the results are expected to be more clustered due to starting conditions (animals of the same type) and experimental design. The human trials are suffering from greater variability of subjects and their individual situations in case of healthy volunteers, whereas reaction of patients is additionally complicated by the stage of disease and specific molecular background. This explains, mentioned several times, complications with numerous nutraceuticals (NTSc) evaluated positively in preclinical studies that fail to demonstrate their efficacy in human clinical trials.

The supplementation with potentially beneficial nutraceutical looks like a good idea, however, there are at least two difficult points: first is the establishing of the susceptible phenotype (several investigated parameters require neural tissue analysis, results are contradictory), whereas the second is related to promising a “miraculous drug” or “miraculous supplement” and financial draining patients and their families. The Authors honestly present their results, unfortunately, their work could be taken out of context and used by unscrupulous sellers. Moreover, the observation that the Nano-PSO works better as a neuroprotectant than a neurorestorative agent may suggest preventive supplementation (before PD diagnosis) with all the consequences for marketing practices. The mentioned shortcomings (lines 742-750) are very important, however, a strong statement is, in my opinion, unavoidable.

There are specific questions resulting from the lecture of the text:

1)       explanation on selection of Nano-PSO instead of regular pomegranate oil, providing more information on Nano-PSO (composition, dose etc.) is required. Reference 73 is a review paper, therefore a specific justification of dose (line 771) should be provided. The dose (800mg/kg/day) seems relatively high when recalculated for human use, especially as the producer suggests daily consumption of two capsulas (weighing each 650 mg, containing 125mg of Pomegranate oil, listed as 100 mg of punicic acid). Selection of this oil should be also justified in the context of the sentence in line 352 (Nano-PSO may thus improve 5HT neurons survival and/or enhance their compensatory functions, as suggested for other omega fatty acids [44].).

2)       line 80: mentioning the numbers of studied animals or patients and comparing them with the current work will be beneficial.

3)       in this work, each group of animals consisted of 40 rats. In specific tests, there are at least five experiments when rats (n=8) were euthanized, then there is a motor experiment (n=10, 4.2. Motor coordination tests) and 4.3. Dopaminergic neuron counts (n=6). Please provide a scheme or description showing which results were obtained from the same animals. As it seems that most phenotypic features were obtained from different groups of rats, finding a common responsive phenotype may be more difficult.

4)       line 185: All rats underwent histological assessments 42 days after starting the experimental interventions. In Methods, the ROT rats were sacrificed after 21 days (line 760). Please clarify this difference.

5)       in line 889, radioimmunoprecipitation buffer is mentioned, which part of the following procedure uses radioimmunoprecipitation?

The figure captions are extremely long, some information is repeated in the text, some details are included in every description.

Captions for particular chapters are very long and convoluted, for example: 2.2. The number of dopaminergic neurons varied markedly among the rat population, regardless of whether they were rotenonized or not and supplemented or not with Nano-PSO.

Comments on the Quality of English Language

In several places, style turns overly complicated and makes the text difficult to follow, there are also fragments that lack precision:

1)       line 140: In contrast, motor performance in intact rats varied a bit between specimens and among trials.

2)       line 354: Now, the results obtained for are displayed in

3)       line 596: The reason why Nano-PSO has, relatively speaking, a “modest” effect on AOx-phenotypes is unclear

4)       line 667: were inclined be categorical

5)       line 684: Thus, PD may result from a process of “dysfunctional maturation” not as the consequence of unsuccessful ageing

6)       1.8. Gene expression profiling (wrong number)

Author Response

Reviewer  #1

Comment:

“The manuscript “The benefit of consuming Nano emulsified punicic acid (Nano-PSO) varies across a population of rotenone-induced parkinsonian albino rats: implications for translational therapies for Parkinson's disease” presents research aimed at developing supporting therapy for Parkinson's disease by application of punicic acid in the form of nanoemulsion. The study utilizes rat model, with investigation how rotenone is affecting the neural functions and the Nano-PSO protecting or restoring them. An important idea in this paper is confronting categorical thinking and phenotypic variation in reaction to treatment (in this case, nutriceutical), with conclusions directed at more personal selection of medicine and supplements.

As the Authors state in the abstract, their results revealed that: 1) The toxicity of rotenone or the benefits of Nano-PSO, the latter used as a neuroprotectant or neurorestorative, varied among rotenonized rats in ranges that went from specimens that showed substantial damage or benefits to those that were marginally affected or benefited. 2) They did not identify any motor, morphological or metabolic parameter or combination of them, which could be used as a reliable index to monitor the deterioration or improvement of the neurometabolic state in the group of rotenonized rats studied.

Several times in the results the Authors describe potential reaction to the test results according to the classical drug development approach “Thus, ruling out the use of a Nano-PSO in phase I clinical studies because the therapeutic response is not significant for the population evaluated, is a mistake, since it reduces the possibilities of many patients of obtaining the benefits of it”. In another fragment, “From a clinical point of view, as a thought experiment, assuming this is a phase one clinical trial, the most receptive subjects could be chosen to continue supplementing Nano-PSO, even if the therapeutic response of the population was poor and would not reach values of significance if parametric methods are used to analyze the data. In our case, nonetheless, numerous neuro-protected rats obtained a great benefit. This was not the case for the neuro-restored rats.”

The suggested benefit for certain phenotypes agrees with postulates of precision medicine and personalized medicine, as the Authors state at the end of Introduction.

The idea of personalized medicine is gaining importance with the progress in studying molecular mechanisms of drug action and subtle differences between individuals. In the manuscript the Authors examined several parameters to assess the effects of rotenone and Nano-PSO, and used hierarchical clustering dendrogram analysis (HCDA) to investigate the phenotyping similarities.

Authors’ response:

We greatly appreciate the initial comments made by Reviewer #1 regarding the intention behind our study. We acknowledge that our original text may not have adequately conveyed this intention, potentially leading to a misunderstanding of our manuscript’s goal. While our study has implications for translating animal experimental treatments to human clinical settings, it was not intended to propose Nano-PSO as a supporting therapy for Parkinson’s disease or to advocate its use for such cases. Instead, our goal was to demonstrate that categorical thinking may undermine the potential use of nutraceuticals (and other pharmaceuticals) as co-adjuvants for preventing and/or treating neurodegeneration. We postulate that this shortcoming can be addressed by incorporating the study of interindividual variance through non-categorical analyses using diverse parameters to monitor improvement (or lack thereof). Such a screening approach could help identify individuals who may benefit, even if the therapeutic response across the population is not uniformly optimal. This aligns with the objectives of personalized medicine. In this context, Nano-PSO and rotenone-treated rats serve as tools to validate the conceptual framework of our work. Accordingly, we have retitled and rewritten the entire manuscript (highlighted in yellow) to make these ideas more explicit, clearer, and easier to understand. We also avoid making disproportionate references and affirmations regarding the utility of our results to human clinical settings. We hope that with all these considerations, Reviewer #1 finds our contribution balanced and suitable for publication.

Reviewer #1

Comment:

However, the results presented in the manuscript show limited effects of Nano-PSO and irregular reactions and relations in investigated groups. It is believed that animal studies offer more controlled conditions, starting from genetic make-up of animals, and the results are expected to be more clustered due to starting conditions (animals of the same type) and experimental design. The human trials are suffering from greater variability of subjects and their individual situations in case of healthy volunteers, whereas reaction of patients is additionally complicated by the stage of disease and specific molecular background. This explains, mentioned several times, complications with numerous nutraceuticals (NTSc) evaluated positively in preclinical studies that fail to demonstrate their efficacy in human clinical trials.

Authors’ response:

As previously commented, we truly believe that our original text may not have conveyed the intention of the manuscript. So, to particularly address this observation, we have clarified further our goal by adding a subsection titled 2.1 General considerations at the beginning of the 2. Results section.

Reviewer #1

Comment:

The supplementation with potentially beneficial nutraceutical looks like a good idea, however, there are at least two difficult points: first is the establishing of the susceptible phenotype (several investigated parameters require neural tissue analysis, results are contradictory), whereas the second is related to promising a “miraculous drug” or “miraculous supplement” and financial draining patients and their families. The Authors honestly present their results, unfortunately, their work could be taken out of context and used by unscrupulous sellers. Moreover, the observation that the Nano-PSO works better as a neuroprotectant than a neurorestorative agent may suggest preventive supplementation (before PD diagnosis) with all the consequences for marketing practices. The mentioned shortcomings (lines 742-750) are very important; however, a strong statement is, in my opinion, unavoidable.

Authors’ response:

Once again, we agree fully with the opinion of R#1. We have introduced this very valuable opinion by rephrasing the last paragraph of the discussion section (lines 737-752) and including an even stronger statement at the end of the conclusion section (lines 767-775).

Reviewer #1

Comments:

There are specific questions resulting from the lecture of the text:

1)       explanation on selection of Nano-PSO instead of regular pomegranate oil, providing more information on Nano-PSO (composition, dose etc.) is required. Reference 73 is a review paper, therefore a specific justification of dose (line 771) should be provided. The dose (800mg/kg/day) seems relatively high when recalculated for human use, especially as the producer suggests daily consumption of two capsulas (weighing each 650 mg, containing 125mg of Pomegranate oil, listed as 100 mg of punicic acid). Selection of this oil should be also justified in the context of the sentence in line 352 (Nano-PSO may thus improve 5HT neurons survival and/or enhance their compensatory functions, as suggested for other omega fatty acids [44].).

2)       line 80: mentioning the numbers of studied animals or patients and comparing them with the current work will be beneficial.

3)       in this work, each group of animals consisted of 40 rats. In specific tests, there are at least five experiments when rats (n=8) were euthanized, then there is a motor experiment (n=10, 4.2. Motor coordination tests) and 4.3. Dopaminergic neuron counts (n=6). Please provide a scheme or description showing which results were obtained from the same animals. As it seems that most phenotypic features were obtained from different groups of rats, finding a common responsive phenotype may be more difficult.

4)       line 185: All rats underwent histological assessments 42 days after starting the experimental interventions. In Methods, the ROT rats were sacrificed after 21 days (line 760). Please clarify this difference.

5)       in line 889, radioimmunoprecipitation buffer is mentioned, which part of the following procedure uses radioimmunoprecipitation?

Authors’ response:

1) We have clarified the dose of Nano-PSO used and justified the dose and the decision of using Nano-PSO as opposed to regular pomegranate oil. We introduced original references to support our calls (lines 788-802). Regarding the “Selection of this oil should be also justified in the context of the sentence in line 352 (Nano-PSO may thus improve 5HT neurons survival and/or enhance their compensatory functions, as suggested for other omega fatty acids [44]”. We have no justification since this finding was unexpected. This was the reason why we just mentioned it, and provided some thoughts on its possible physiological meaning without getting into further details. 

2). We have added the information solicited but rephrased the paragraph to highlight the elements that are important for the manuscript (lines 78-89).

3) We have improved the quality of the writing of the first paragraph of the subsection 4.1 Animals (lines 778-805) in the section of Materials and Methods, and introduced a table titled “experimental design” that specifies which results were obtained from the same animal. The table can be consulted in the supplementary material.

 4) This contradiction has been corrected. All rats used to conduct morphological assessments were sacrificed 42 days following the initiation of treatment.

5) We did not use immunoprecipitation in this work. However, we did use RIPA buffer to homogenize the samples destined to estimate lipid peroxidation as recommended by the supplier: “For Tissue Homogenates: weigh out approximately 25 mg of tissue into a 1.5 mL centrifuge tube.  Add 250 μL of RIPA Buffer containing protease inhibitors of choice (see Interference section).  Homogenize or sonicate the tissue on ice.  Centrifuge the tube at 1,600 x g for 10 minutes at 4°C. Use the supernatant for analysis. Store supernatant on ice. If not assaying the same day, freeze at -80°C. The sample will be stable for one month. Tissue homogenates do not need to be diluted before assaying”. https://www.abnova.com/upload/media/product/protocol_pdf/KA1381.pdf

Reviewer #1

Comments:

The figure captions are extremely long, some information is repeated in the text, some details are included in every description.

Captions for particular chapters are very long and convoluted, for example: 2.2. The number of dopaminergic neurons varied markedly among the rat population, regardless of whether they were rotenonized or not and supplemented or not with Nano-PSO.

Authors’ response:

All figure captions have been simplified and straightened.

Reviewer #1

Comments:

Comments on the Quality of English Language

In several places, style turns overly complicated and makes the text difficult to follow, there are also fragments that lack precision:

1)       line 140: In contrast, motor performance in intact rats varied a bit between specimens and among trials.

2)       line 354: Now, the results obtained for are displayed in

3)    line 596: The reason why Nano-PSO has, relatively speaking, a “modest” effect on AOx-phenotypes is unclear

4)       line 667: were inclined be categorical

5)       line 684: Thus, PD may result from a process of “dysfunctional maturation” not as the consequence of unsuccessful ageing

6)       1.8. Gene expression profiling (wrong number)

Authors’ response:

English style was corrected, and imprecise sentences avoided throughout the entire text.

Reviewer 2 Report

Comments and Suggestions for Authors

Work very comprehensive, well-written introduction. Innovative topic, raising important issues. A number of advanced statistical methods and tests were used. Methodology described in detail. Results appear to be reliable, and aesthetically visualized. The discussion is well led and the conclusions are written based on the results. I see no need for any changes. I recommend for publication in its current form.

Author Response

Reviewer #2

Comment:

1) Work very comprehensive, well-written introduction. Innovative topic, raising important issues. A number of advanced statistical methods and tests were used. Methodology described in detail. Results appear to be reliable, and aesthetically visualized. The discussion is well led and the conclusions are written based on the results. I see no need for any changes. I recommend for publication in its current form.

Authors’ response:

We thank Reviewer’s #2 considerations. Still, we believe our review enhanced the quality of the manuscript. Hopefully, she/he agrees with our belief.

Reviewer 3 Report

Comments and Suggestions for Authors

The paper presents interesting information. On the whole, manuscript is well prepared, high-quality studies are presented. Please see some sugesstions:

Abstract: please add more detail information about obtained results

Introduction: the last part ( lines88-111) should be shortened. Most of the information should be included in conclusions.

Results: well described along with good quality figures. Please improve line 252: remove dot (α-.synuclein..)

Discussion: please add references to lines 653-673, 725-740

Methods: all information are provided, methods are adequate to aim of the studies

Statistics: provided

Conclusions: lack, please provide

References: up-to-date, but a few positions should be added : see Discussion

Author Response

Reviewer #3

Comments:

1) Abstract: please add more detail information about obtained results

2) Introduction: the last part ( lines88-111) should be shortened. Most of the information should be included in conclusions.

3) Results: well described along with good quality figures. Please improve line 252: remove dot (α-.synuclein..)

4) Discussion: please add references to lines 653-673, 725-740

5) Methods: all information are provided, methods are adequate to aim of the studies

6) Statistics: provided

7) Conclusions: lack, please provide

8) References: up-to-date, but a few positions should be added : see Discussion

Authors’ response:

We attended all the of the Reviewer’s #3 suggestions. We hope we have done correctly.